# PAC-Bayes bounds for cumulative loss in Continual Learning

**Lior Friedman**      **Ron Meir**
Department of Electrical Engineering
Technion Institute of Technology
Haifa, Israel
`liorf@campus.technion.ac.il, rmeir@ee.technion.ac.il`

## Abstract

In continual learning, knowledge must be preserved and re-used between tasks, requiring a balance between maintaining good transfer to future tasks and minimizing forgetting of previously learned ones. As several practical algorithms have been devised to address the continual learning setting, the natural question of providing reliable risk certificates has also been raised. Although there are results for specific settings and algorithms on the behavior of memory stability, generally applicable upper bounds on learning plasticity are few and far between.

In this work, we extend existing PAC-Bayes bounds for online learning and time-uniform offline learning to the continual learning setting. We derive general upper bounds on the cumulative generalization loss applicable for any task distribution and learning algorithm as well as oracle bounds for Gibbs posteriors and compare their effectiveness for several different task distributions. We demonstrate empirically that our approach yields non-vacuous bounds for several continual learning problems in vision, as well as tight oracle bounds on linear regression tasks. To the best of our knowledge, this is the first general upper bound on learning plasticity for continual learning.

## 1 Introduction

Continual learning is a machine learning setting in which collections of examples, known as tasks, arrive sequentially. These tasks may be different skills and capabilities, represent changes in the data distribution over time, or encapsulate different contexts or environments. Since tasks change over time, continual learning is also referred to as incremental learning or lifelong learning. Due to the limited model capacity and the sequential nature of the continual learning setting, issues often arise in adapting the model to new tasks as they appear while also preserving its performance on previous tasks and thus avoiding *Catastrophic Forgetting* (Goodfellow et al., 2015; Kirkpatrick et al., 2017), where performance on previous tasks degrades significantly as the model adapts to new tasks. This dilemma is a facet of the trade-off between learning plasticity and memory stability (Wang et al., 2024a), two key aspects of the learner's behavior during the continual learning process.

There are several methods and algorithms aiming to effectively resolve this tradeoff in various continual learning scenarios (see Wang et al. (2024a)), such as via regularization (Kirkpatrick et al., 2017), replay of data from previous tasks (Rebuffi et al., 2017) or other methods. Due to the inherent trade-off between forgetting and forward transfer, metrics to evaluate continual learning algorithms differ. Common metrics include the average accuracy of the model for all previous tasks, memory stability measures such as forgetting, and learning plasticity measures such as intransigence (Chaudhry et al., 2018), forward transfer and cumulative error incurred in each task.

While several empirical methods aimed to tackle the challenges of continual learning have been proposed in recent years, there are significantly fewer theoretical works aiming to analyze the properties of continual learning problems and provide estimates or guarantees on overall performance. Some works (Evron et al., 2022; Doan et al., 2021; Benavides-Prado & Riddle, 2022) focus on understanding the behavior of forgetting, an interesting topic in and of itself, while other works (Lin et al., 2023; Li et al., 2024; Bennani & Sugiyama, 2020; Zhao et al., 2024; Levinstein et al., 2025)

focus on average model loss for specific settings such as continual linear regression or the Neural Tangent Kernel (NTK) regime (e.g. very wide neural networks). One promising direction to derive more general performance guarantees is the PAC-Bayes framework (McAllester, 1999; Catoni, 2004; Alquier, 2024). Existing upper bounds on meta-learning (Pentina & Lampert, 2015; Amit & Meir, 2018; Balcan et al., 2019; Chen et al., 2023) and online learning (Haddouche & Guedj, 2022) offer guarantees for the loss on unseen tasks as well as the online setting that serves as a special case of continual learning. A recent paper by Friedman & Meir (2025) derived general PAC-Bayes bounds on forgetting in the continual learning setting. To the best of our knowledge, however, there are no PAC-Bayes bounds for the cumulative loss in the continual learning setting. The cumulative loss and plasticity in general has received a surge of interest in recent years (Wang et al., 2024b; Dohare et al., 2024; Kumar et al., 2025).

In this work, we extend existing PAC-Bayes bounds for online learning (Haddouche & Guedj, 2022) and time-uniform offline learning (Haddouche & Guedj, 2023; Chugg et al., 2023) to the continual learning setting, allowing us to derive the first (to our knowledge) algorithm-agnostic risk certificates for learning plasticity via the cumulative loss. We derive bounds applicable for bounded and sub-Gaussian losses for offline and online continual learning. Equation 3, for instance, suggests that if we continually re-train a model daily over a year, only a few dozen examples are required to provide effective high-probability risk certificates for the expected cumulative loss. We also analyze our bounds as oracle bounds and compare their effectiveness for several different task distributions to provide additional insights into the relationship between forward transfer, model complexity and task similarity. We demonstrate empirically that our approach yields non-vacuous bounds for several continual learning problems in vision, as well as tight oracle bounds on linear regression tasks.

## 2 BACKGROUND

### 2.1 PAC-BAYES BOUNDS

We provide here a brief, high-level overview of several key notions related to PAC-Bayes bounds. For a more complete introduction including detailed proofs, see Alquier (2024). In the classical supervised learning setting, the learner attempts to learn an unknown data distribution $\mathcal{D}$ via a training dataset of size $m$, $S = \{z_1, \ldots, z_m\}$ where $z_i \in \mathcal{Z}$. A standard assumption is that the training dataset is sampled i.i.d. from $\mathcal{D}$. A learning algorithm assigns a probability to each hypothesis $h$ in hypothesis class $\mathcal{H}$ (a deterministic algorithm selects a single hypothesis). A loss function $\ell : Z \to \mathbb{R}^+ \cup \{0\}$ (e.g. classification error) is used to measure the performance of a learning algorithm. Since $\mathcal{D}$ is unknown, the expected risk $\mathcal{L}(h, \mathcal{D}) \triangleq \mathbb{E}_{z \sim \mathcal{D}} \ell(h, z)$ cannot be minimized directly, and so a generalization bound would upper bound the gap between it and the empirical loss $\hat{\mathcal{L}}(h, S) = \frac{1}{m} \sum_{i=1}^{m} \ell(h, z_i)$. PAC-Bayes bounds provide an upper bound on this gap with high probability w.r.t. the empirical sample $S$. A classic example is Catoni's bound (Catoni, 2004)

**Theorem 2.1.** *(Catoni's single task bound) Assume* $\forall h \in \mathcal{H}, z \in \mathcal{D}, \ell(h, z) \in [0, K]$. *Let* $P \in \mathcal{M}(\mathcal{H})$ *be some data-free prior distribution over* $\mathcal{H}$. *Then, for any* $\lambda > 0$, *for any* $\delta \in (0, 1)$, *with probability at least* $1 - \delta$ *over the choice of* $S$, *uniformly for all posteriors* $Q \in \mathcal{M}(\mathcal{H})$,

$$\mathcal{L}(Q, \mathcal{D}) \leq \hat{\mathcal{L}}(Q, S) + \frac{1}{\lambda} D_{KL}(Q \| P) + \frac{\lambda K^2}{8m} + \frac{\log 1/\delta}{\lambda} .$$

Of note is that this upper bound applies uniformly over all posteriors, implying that we can optimize the r.h.s. w.r.t. the posterior $Q$, and that the bound strongly depends on a given data-free prior. Recent work such as Pérez-Ortiz et al. (2021) showed that using a part of the training set to learn the prior can lead to tighter bounds in practice. Notably, all of the training data may be used for finding the posterior $Q$. While we will not go over the full proof structure for Catoni's bound, we note that there are three main components: (1) Markov's inequality (2) A change-of-measure inequality e.g. (Donsker & Varadhan, 1975) (3) Hoeffding's inequality or extensions thereof. These components or some variation of them are common elements used for most PAC-Bayes bounds.

### 2.2 CONTINUAL LEARNING AND PAC-BAYES BOUNDS

In order to obtain upper bounds on the cumulative loss for continual learning, we must first define a sufficient theoretical framework for describing the learning process in question. Figure 1 provides

an overview of the continual learning process using the same symbolic terminology described below. We follow a similar framework as Haddouche & Guedj (2022; 2023), adapted to the continual

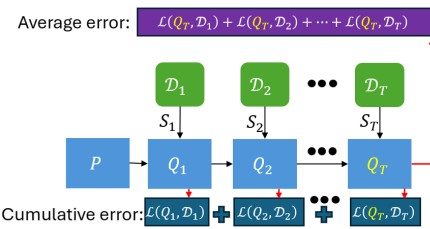

Figure 1: Depiction of the continual learning process. A data-free prior $P$ is adapted to task $\mathcal{D}_1$ via an empirical sample $S_1 \sim \mathcal{D}_1^m$, resulting in posterior $Q_1$. The posterior's expected (test) loss $\mathcal{L}(Q_1, \mathcal{D}_1)$ is added to the cumulative loss. $Q_1$ serves as the new prior for the next task $\mathcal{D}_2$ and so on until we reach a final posterior $Q_T$. The cumulative loss is the sum of errors for all tasks.

learning setting by changing each task from a single example to a set of i.i.d. samples.

**Framework**  Consider a data space $\mathcal{Z} = \mathcal{X} \times \mathcal{Y}$ defined as pairs or inputs and outputs, i.e. $z_i = (x_i, y_i) \in \mathcal{Z}$, where $x_i \in \mathcal{X}, y_i \in \mathcal{Y}$. We fix an integer sample size $m > 0$. For our continual learning process, we consider a finite sequence of tasks of length $T$, where each task $t \in [T]$ is associated with an unknown data distribution $\mathcal{D}_t$. We assume that for each task $t$ we are given an i.i.d. sample of size $m$ from the corresponding data distribution, $S_t \in \mathcal{D}_t^m$. Our complete data sample is therefore composed of the sequence $(S_t)_{t=1}^T \triangleq (S_1, S_2, \ldots, S_T)$. We make no assumptions regarding the relation between tasks or the length of the task sequence, and our results are applicable for any $T > 0$. We denote our hypothesis space $\mathcal{H}$ such that $h \in \mathcal{H}$ is a function $h : \mathcal{X} \to \mathcal{Y}$ and denote the space of probability distributions over $\mathcal{H}$ as $\mathcal{M}_1(\mathcal{H})$.

We set a sequence of priors, starting with a data-free prior distribution $P_1 = P \in \mathcal{M}_1(\mathcal{H})$, and $(P_t)_{t \geq 2}$ such that each $P_t \in \mathcal{M}_1(\mathcal{H})$ is $\mathcal{F}_{t-1}$-measurable, with $(\mathcal{F}_t)_{t \geq 0}$ being an adapted filtration to $(S_t)_{t=1}^T$. We denote $(Q_t)_{t=1}^T$ a sequence of posterior distributions such that $Q_t$ is absolutely continuous w.r.t. $P_t$. Similarly to Haddouche & Guedj (2022), we introduce the notion of stochastic kernels (Rivasplata et al., 2020) as data-dependent measures within the PAC-Bayes framework, allowing us to define bounds with data-dependent priors. We denote $\Sigma_{\mathcal{H}}$ the $\sigma$-algebra on $\mathcal{H}$.

**Definition 1.** (*Stochastic Kernels*) A stochastic kernel from $S \in \mathcal{Z}^m$ to $\mathcal{H}$ is defined as a mapping $Q : \mathcal{Z}^m \times \Sigma_{\mathcal{H}} \to [0, 1]$ where (1) For any $B \in \Sigma_{\mathcal{H}}$, the function $S \to Q(S, B)$ is measurable. (2) For any $S \in \mathcal{Z}^m$, the function $B \to Q(S, B)$ is a probability measure over $\mathcal{H}$.

Stochastic kernels allow us to refer to a distribution dependent on dataset $S$, marking $Q_S = Q(S, \cdot)$.

**Definition 2.** A sequence of stochastic kernels $(P_t)_{t=1}^T$ is called an *online predictive sequence* if (1) For all $t \geq 1$, $S \in \mathcal{Z}^m$, $P_t(S, \cdot)$ is $\mathcal{F}_{t-1}$-measurable and (2) for all $t \geq 2$, $P_t(S, \cdot)$ is absolutely continuous w.r.t. $P_{t-1}(S, \cdot)$. We will also use the notation $P_{1:t}$ to denote such a kernel.

This definition of online predictive sequences allows for the description of a stochastic kernel (and posterior distribution for hypotheses) that changes as new tasks arrive sequentially, and can be used to describe a variety of continual learning algorithms.

**Definition 3.** A loss function $\ell : \mathcal{H} \times \mathcal{Z} \to \mathbb{R}^+ \cup \{0\}$ is a function mapping a hypothesis $h$ and data sample $z$ to the set of nonnegative numbers. The expected loss for $h \in \mathcal{H}$ is $\mathcal{L}(h, \mathcal{D}) \triangleq \mathbb{E}_{z \in \mathcal{D}} \ell(h, z)$. The empirical loss of a hypothesis w.r.t. a sample $S \in \mathcal{Z}^m$ is $\hat{\mathcal{L}}(h, S) \triangleq \frac{1}{m} \sum_{j=1}^m \ell(h, z_j)$.

We assume for the sake of convenience that the loss function $\ell(h, z)$ is task-agnostic, but our results are applicable even if the loss function depends on the task identifier. We restate the main result of Haddouche & Guedj (2022) for online learning using our terminology in Theorem A.1. Finally, we define the cumulative loss (CuL) and the average loss (AL). We note that this paper will focus on the cumulative loss, corresponding to notions such as forward transfer (FWT), intransigence and learning plasticity rather than the average loss (corresponding to average accuracy). For a more indepth overview of these metrics, see Wang et al. (2024a). We note that both metrics refer to expected (test) errors.

**Definition 4.** For a given online predictive sequence $(Q_t)$, a sequence of tasks $\mathcal{D}_1, \dots, \mathcal{D}_T$, and a sample $S_{1:T} = (S_1, \dots, S_T) \sim \mathcal{D}_1^m \times \dots \times \mathcal{D}_T^m$, the *cumulative loss (CuL)* is defined as

$$\mathrm{CuL}((Q_t)_{t=1}^T) = \sum_{t=1}^T [\mathcal{L}(Q_{t,S_{1:t}}, \mathcal{D}_t)|\mathcal{F}_{t-1}] \triangleq \sum_{t=1}^T \mathbb{E}_{h_t \sim Q_{t,S_{1:t}}}[\mathbb{E}_{z_t \sim \mathcal{D}_t}[\ell(h_t, z_t)|\mathcal{F}_{t-1}]].$$

The *Average loss (AL)* is defined as

$$\mathrm{AL}(Q_T) = \sum_{t=1}^T \mathcal{L}(Q_{T,S_{1:T}}, \mathcal{D}_t) \triangleq \sum_{t=1}^T \mathbb{E}_{h_T \sim Q_{T,S_{1:T}}}[\mathbb{E}_{z_t \sim \mathcal{D}_t}[\ell(h_T, z_t)]].$$

As we can see from the definitions, the CuL is measured w.r.t. the entire predictive sequence sequentially, and can be extended to new tasks by simply adding an additional element for task $T + 1$. The AL, on the other hand, is measured retroactively for a single posterior w.r.t. all tasks, must be re-calculated from scratch for each new task, and requires additional memory proportional to the number of tasks in order to estimate the expected loss for each task. We note that both terms differ from the meta-learning loss, that can be expressed (for unknown task $\mathcal{D}_{T+1}$) as $\mathrm{MetaL}(Q_T) \triangleq \mathbb{E}_{S_{T+1} \sim \mathcal{D}_{T+1}^m}[\mathcal{L}(Q_T(S_{1:T+1}), \mathcal{D}_{t+1})]$. Unlike CuL and AL, the meta-learning loss considers the loss on a future task rather than past or current performance. We note that this framework allows for the derivation of risk certificates for both offline continual learning, where samples for each task are given as a batch, and online continual learning, where each sample is given sequentially and cannot be re-used.

## 2.3 OTHER RELATED WORK

**Continual learning** In recent years, there have been several prominent papers focused on understanding forgetting and average error in the context of continual learning. Evron et al. (2022) as well as Lin et al. (2023) and Li et al. (2024) provide upper bounds and equations that describe expected behavior for sets of noisy linear regression tasks. Factors such as the number of parameters vs the number of samples, task relationships and task order are all relevant parts of forgetting. Other works such as Bennani & Sugiyama (2020) and Doan et al. (2021) provide upper bounds on average and cumulative errors for the Stochastic Gradient Descent (SGD) algorithm as well as the Orthogonal Gradient Descent (OGD) (Farajtabar et al., 2020) algorithm, for models that correspond to the Neural Tangent Kernel (NTK) regime. Friedman & Meir (2025) provide general PAC-Bayes upper bounds on average error that are applicable regardless of optimization method, task structure and hypothesis class for bounded loss functions. Our bounds instead focus on the cumulative loss, a measure of learning plasticity. Unlike results for the NTK regime, our bounds are algorithm and architecture agnostic.

**Meta-learning** There are numerous similarities between continual and meta-learning. We note that in meta-learning data from previous tasks remains accessible, and tasks are usually assumed to be taken from a single task-generating distribution. Pentina & Lampert (2015) as well as Amit & Meir (2018) derive PAC-Bayes bounds for the meta-learning loss described above for bounded loss functions. Other relevant works consider the cumulative loss and overall regret in the meta-learning problem from a convex optimization perspective; Balcan et al. (2019) show upper bounds for variants of the online SGD algorithm and follow-the-leader methods under some assumptions on loss convexity and the parameter space. We note that follow-the-leader methods require access to data from previous task, making their application to continual learning problems difficult.

**Continual Meta-learning** Chen et al. (2023) consider the problem of continual meta-learning, where a meta-parameter is updated based on past and current task data and task-specific parameters can only access data from the current task. For loss functions bounded in $[0, 1]$ and several base-learner algorithms, they derive upper bound on excess risk incurred in both the static and shifting environments. Interestingly, the meta-algorithm used does not require data from past tasks, allowing its application in continual learning, unlike follow-the-leader methods. In contrast to the above methods, our general bounds are applicable for both offline and online continual learning, regardless of optimization algorithm, model architecture and task environment.

## 3 UPPER BOUNDS ON CUMULATIVE LOSS

**Assumption 1.** The loss function $\ell(h, z)$ is either: (1) upper bounded by constant $K$ or (2) is $K$ sub-Gaussian. Formally, $\forall h \in \mathcal{H}, z \in \mathcal{Z}, \ell(h, z) \in [0, K]$ or $\mathbb{E}_{z \sim \mathcal{D}_i}\left[e^{t(\ell(h,z) - \mathbb{E}_{z \sim \mathcal{D}_i} \ell(h,z))}\right] \leq e^{Kt^2}$.

We note that Assumption 1 only applies to the loss used for the upper bound calculation. Some relevant examples include classification loss, cross entropy loss with non-zero minimal class probability, and any light-tailed unbounded loss. First, we extend the result of Theorem A.1 to the continual learning setting ($m > 1$) and to sub-Gaussian losses.

**Corollary 3.1.** *Under Assumption 1, for any $\lambda > 0$, for any set of $T$ tasks, for any online predictive sequence of priors $(P_t)$, for any sequence of stochastic kernels $(Q_t)$ with probability at least $1 - \delta$ over the draw of $(S_1, \ldots, S_T) \sim \mathcal{D}_1^m \times \ldots \times \mathcal{D}_T^m$, we have that the following holds for the data-dependent measures $P_t(S_{1:t}, \cdot), Q_t(S_{1:t}, \cdot)$,*

$$\frac{1}{T}\text{CuL}((Q_t)_{t=1}^T) \leq \frac{1}{T} \sum_{t=1}^T \hat{\mathcal{L}}(Q_t(S_{1:t}), S_t) + \frac{1}{\lambda T} \sum_{t=1}^T \text{KL}(Q_t(S_{1:t}) || P_t(S_{1:t}))$$
$$+ \frac{\lambda K^2}{m} + \frac{\log(1/\delta)}{\lambda T}. \tag{1}$$

The full proof is in Appendix A, and follows a similar outline to Theorem A.1 but adapted to the continual learning setting. In particular, unlike the online setting, if we have $m \gg T$, the r.h.s. converges to the l.h.s. as $m \to \infty$, similarly to the standard PAC-Bayes bounds. This result is a straightforward extension of Theorem A.1 and requires no new technical tools.

Since the prior sequence may be data-dependent, we can select $P_1 = P$ and $P_t(S_{1:t}) = Q_{t-1}(S_{1:t})$, similarly to Haddouche & Guedj (2022), giving us a KL-divergence term that depends on the change in the posterior between subsequent tasks. We note that while the posterior stochastic kernel $Q_{t-1}$ must be $\mathcal{F}_{t-2}$-measurable and is therefore independent of $S_t$, the measure $Q_{t-1}(S_{1:t}, \cdot)$ can use $S_t$ in equation 1. In practical terms, this means that for models with task-specific parameters (i.e. parameters that are used only for specific tasks), KL-divergence can be measured w.r.t. only shared parameters. This may serve to lower the overall KL-divergence, as task-specific parameters can be used to specialize a more general model that varies little between tasks.

While Corollary 3.1 is a useful upper bound, equation 1 has terms in the r.h.s. that do not converge to the l.h.s. as the number of tasks $T$ increases, namely the term $\lambda K^2/m$. In the context of continual learning, upper bounds that converge as the number of tasks increases are vastly preferable as they converge even if the number of samples per task is fixed. In order to achieve such bounds, we must make some additional assumptions on our tasks. Specifically, we assume that the loss function is upper bounded by a constant, and that the number of samples per task is not much smaller than the number of tasks, i.e. $m \gg \sqrt{T}$.

**Theorem 3.2.** *Under the same setup as Corollary 3.1, assuming $\ell(h, z) \in [0, K]$, for any predictive sequence of posteriors $(Q_t)$, for any $\delta_2 \in (0, 1]$, with probability at least $1 - \delta$ over the draw of $(S_1, \ldots, S_T) \sim \mathcal{D}_1^m \times \ldots \times \mathcal{D}_T^m$, the following holds for measures $P_t(S_{1:t}), Q_t(S_{1:t})$*

$$\frac{1}{T}\text{CuL}((Q_t)_{t=1}^T) \leq \frac{1}{T} \sum_{t=1}^T \hat{\mathcal{L}}(Q_t(S_{1:t}), S_t) + \frac{1}{\lambda} \sum_{t=1}^T \text{KL}(Q_t(S_{1:t}) || P_t(S_{1:t}))$$
$$+ \frac{1}{\lambda} \log\left\{(1 - \delta_2)e^{\lambda K \sqrt{\log(1/\delta_2)/2mT}} + \delta_2 e^{\lambda K}\right\} + \frac{\log(1/\delta)}{\lambda}. \tag{2}$$

Specifically, we can set $\lambda = \frac{T\sqrt{T}}{K}, \delta_2 = e^{-T\sqrt{T}}$, resulting in

$$\frac{1}{T}\text{CuL}((Q_t)_{t=1}^T) \leq \frac{1}{T} \sum_{t=1}^T \hat{\mathcal{L}}(Q_t(S_{1:t}), S_t) + \frac{K}{T\sqrt{T}} \sum_{t=1}^T \text{KL}(Q_t(S_{1:t}) || Q_{t-1}(S_{1:t}))$$
$$+ \frac{K\sqrt[4]{T}}{\sqrt{2m}} + \frac{K(1 + \log(1/\delta))}{T\sqrt{T}}, \tag{3}$$

where $Q_0 = P$, meaning that as long as $m > \sqrt{T}/2$ the r.h.s. converges to the l.h.s. as $m, T \to \infty$. This result gives us an upper bound that converges as more tasks are added, provided that we have sufficient samples per task. It also suggests a good rule of thumb for effective continual learning - the number of samples per task should exceed the square root of the number of total tasks.

Proof of Theorem 3.2 appears in Appendix A. This Theorem uses a careful analysis of task and posterior dependencies that is specific to online and continual learning, as well as bad-event analysis that is not commonly utilized in the context of PAC-Bayes bounds.

## 4 ORACLE BOUNDS FOR SPECIFIC SETTINGS

Corollary 3.1 and Theorem 3.2 provide general upper bounds that are applicable for a wide variety of tasks and continual learning algorithms, but their generality makes detailed analysis more difficult. In order to better demonstrate these results and compare them for different settings, we derive and compare oracle upper bounds on the Gibbs posterior.

**Definition 5.** Recall the *Gibbs posterior* distribution and the *expected Gibbs posterior* distribution, for $t \geq 1, \lambda > 0$, for $\hat{Q}_0^\lambda = Q_0^\lambda = P$,

$$\hat{Q}_t^\lambda(h) \propto e^{-\lambda \hat{L}(h, S_t)} \hat{Q}_{t-1}^\lambda(h), \quad Q_t^\lambda(h) \propto e^{-\lambda \mathbb{E}_{z_t \sim \mathcal{D}_t} \ell(h, z_t)} Q_{t-1}^\lambda(h).$$

The main advantage of the Gibbs posterior is that it removes the KL-divergence terms from the r.h.s. of equation 1 (as can be seen, for example, in Lemma A.4). This property allows us to derive upper bounds on the cumulative loss for the sequence of expected Gibbs posteriors. We note that while these bounds refer to the expected Gibbs posterior and discuss the limit where $m, T \to \infty$, they originate from a non-asymptotic bound appearing in Appendix A. Equivalent bounds for empirical Gibbs posteriors can be derived by changing the final condition to be taken w.r.t. the empirical loss on the training set.

**Theorem 4.1.** *For any $\lambda > 0$, assuming (1) $Q_t = Q_t^\lambda$ is the expected Gibbs posterior. (2) $P_1 = P$ is a data-free measure over $\mathcal{H}$. (3) $\forall t > 1 : P_t = Q_{t-1} = Q_{t-1}^\lambda$ (4) $\mathcal{H}$ is a compact, bounded subset of $\mathbb{R}^d$. (5) $\forall t \in [2, T]$, the total expected loss $\sum_{i=1}^t \mathcal{L}(h, \mathcal{D}_i)$ has a strict global minimum at $h_{1:t}^*$ and is twice continuously differentiable w.r.t. $h$. Under Assumption 1, we have*

$$\lim_{m, T \to \infty} \frac{1}{T} \text{CuL}((Q_t^\lambda)_{t=1}^T) \leq \lim_{T \to \infty} \frac{1}{T} \sum_{t=2}^T \mathcal{L}(h_{1:t-1}^*, \mathcal{D}_t). \tag{4}$$

The full proof is provided in Appendix A and relies mainly on Laplace's method (Hwang, 1980; Shun & McCullagh, 1995), which is rarely used for PAC-Bayes bounds. We note that under different strict minimum assumptions we can derive another useful oracle bound,

**Corollary 4.2.** *For any $\lambda > 0$, assuming (1) $Q_t = Q_t^\lambda$ is the expected Gibbs posterior. (2) $P_1 = P$ is a data-free measure over $\mathcal{H}$ (3) $\forall t > 1 : P_t = Q_{t-1} = Q_{t-1}^\lambda$ (4) $\mathcal{H}$ is a compact, bounded subset of $\mathbb{R}^d$. (5) $\forall t \in [2, T]$, the expected loss $\mathcal{L}(h, \mathcal{D}_{t-1})$ has a strict global minimum at $h_{t-1}^*$ and is twice continuously differentiable w.r.t. $h$. Under Assumption 1, we have*

$$\lim_{m \to \infty} \frac{1}{T} \text{CuL}((Q_t^\lambda)_{t=1}^T) \leq \frac{1}{T} \sum_{t=2}^T \mathcal{L}(h_{t-1}^*, \mathcal{D}_t) + \frac{1}{T} \mathcal{L}(P, \mathcal{D}_1). \tag{5}$$

In both cases, we see that the CuL can be upper bounded by the loss of a predictor obtained from previous tasks. Using these oracle bounds, we can compare the cumulative loss for several different continual learning setups and contrast the effects of task similarity between them.

**Assumption 2.** (Lipchitz loss) For non-negative $\mathcal{G}_\mathcal{H}$ and metric $d(\cdot, \cdot)$ (e.g. Wasserstein distance),

$$\forall i, j, h \in \mathcal{H}, \quad |\mathcal{L}(h, \mathcal{D}_i) - \mathcal{L}(h, \mathcal{D}_j)| \leq \mathcal{G}_\mathcal{H} d(\mathcal{D}_i, \mathcal{D}_j).$$

This assumption implies that similarities in the task distribution are reflected in the loss function.

**Assumption 3.** For each task $\mathcal{D}_t$, the optimal (minimum) loss $\mathcal{L}_t^*$ is a achievable via a hypothesis of dimension $d_t$. Additionally, for any two tasks $\mathcal{D}_i, \mathcal{D}_j$, if $d = \dim(\mathcal{H}) \geq d_i + d_j$, i.e. $\mathcal{H}$ is over-parametrized, then there exists $h_{i,j}^* \in \mathcal{H} \subseteq \mathbb{R}^d$ such that $\mathcal{L}(h_{i,j}^*) = \min_{h \in \mathcal{H}} (\mathcal{L}(h, \mathcal{D}_i) + \mathcal{L}(h, \mathcal{D}_j))$.

This assumption aims to link model complexity and representation power, stating that given enough parameters, it is possible to find a hypothesis achieving minimal loss and that given more parameters, we can find optimal hypotheses on the average loss. Taking into account all of our assumptions, we

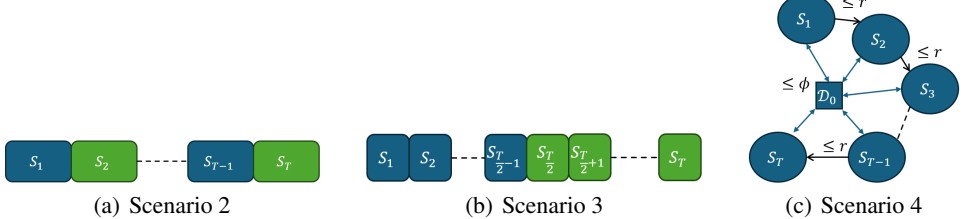

(a) Scenario 2     (b) Scenario 3     (c) Scenario 4

Figure 2: Depictions of scenarios 2, 3 and 4. In scenario 2 we alternate between two task distributions. In scenario 3 we take $T/2$ tasks from one distribution and another $T/2$ from a second one. In scenario 3, each new task is of distance at most $r$ from the previous one, and any two tasks are at distance at most $2\phi$ from each other.

can provide oracle bounds for several different scenarios in terms of task similarities and ordering.

1. If $\forall t, \quad \mathcal{D}_t = \mathcal{D}_1$, i.e. we have different samples from the same task distribution, we have

$$\lim_{m \to \infty} \frac{1}{T} \mathrm{CuL}((Q_t^\lambda)_{t=1}^T) \le \mathcal{L}_1^* + O\left(\frac{1}{T}\right).$$

2. If tasks alternate between distributions $\mathcal{D}_1$ and $\mathcal{D}_2$, we have that if $d \ge d_1 + d_2$, we also obtain the optimal (minimal) result

$$\lim_{m \to \infty} \frac{1}{T} \mathrm{CuL}((Q_t^\lambda)_{t=1}^T) \le \frac{\mathcal{L}_1^* + \mathcal{L}_2^*}{2} + O\left(\frac{1}{T}\right),$$

and otherwise we have (via Theorem 4.1)

$$\lim_{m \to \infty} \frac{1}{T} \mathrm{CuL}((Q_t^\lambda)_{t=1}^T) \le \frac{\min_h \{\mathcal{L}(h, \mathcal{D}_1) + \mathcal{L}(h, \mathcal{D}_2)\}}{2} + \frac{1}{2}\mathcal{G}_\mathcal{H} d(\mathcal{D}_1, \mathcal{D}_2) + O\left(\frac{1}{T}\right)$$

$$\le \min\{\mathcal{L}_1^*, \mathcal{L}_2^*\} + \mathcal{G}_\mathcal{H} d(\mathcal{D}_1, \mathcal{D}_2) + O\left(\frac{1}{T}\right).$$

3. If the first $T/2$ tasks are from $\mathcal{D}_1$ and the second half are from $\mathcal{D}_2$, we have

$$\lim_{m \to \infty} \frac{1}{T} \mathrm{CuL}((Q_t^\lambda)_{t=1}^T) \le \begin{cases} \frac{\mathcal{L}_1^* + \mathcal{L}_2^*}{2} + O\left(\frac{1}{T}\right) & d \ge d_1 + d_2 \\ \frac{\mathcal{L}_1^* + \mathcal{L}_2^*}{2} + O\left(\frac{1}{T}\right) + \frac{1}{T}\mathcal{G}_\mathcal{H} d(\mathcal{D}_1, \mathcal{D}_2) & d < d_1 + d_2 \end{cases}$$

4. If tasks change gradually and do not differ significantly from one another, or more formally, if $\forall t, \quad d(\mathcal{D}_t, \mathcal{D}_0) \le \phi$ and $\forall t < T, \quad d(\mathcal{D}_t, \mathcal{D}_{t+1}) \le r, r < \phi$, we have

$$\lim_{m \to \infty} \frac{1}{T} \mathrm{CuL}((Q_t^\lambda)_{t=1}^T) \le \frac{1}{T} \sum_{t=1}^T \mathcal{L}_t^* + r\mathcal{G}_\mathcal{H} + O\left(\frac{1}{T}\right).$$

These results imply that (given limited capacity) task order can significantly impact the cumulative error. This is not particularly surprising, as existing results on the expected loss and forgetting in continual learning also demonstrate that task order and relatedness are a major factor in overall error, but it is not immediate to deduce that this should also be the case for the cumulative loss. These results also demonstrate that sufficiently over-parametrized models can overcome issues such as task order and learn each task effectively.

## 5 EMPIRICAL DEMONSTRATION

In this section we demonstrate the utility of our bounds from Sections 3 and 4. We study the efficacy of our bounds several simple (but non-trivial) environments with varied task similarity.

Table 1: Average and final cumulative error percentage for vision tasks. Lower is better.

| Domain | Method | CuL | Bound (equation 3) | Error @$t = 120$ | Bound @$t = 120$ |
|--------|--------|-----|--------------------|------------------|------------------|
| Perm.-MNIST | EWC | $1.0 \pm 0.0$ | $10.6 \pm 0.2$ | $1.0 \pm 0.0$ | $5.1 \pm 0.2$ |
| | VI | $15.5 \pm 0.1$ | $17.9 \pm 0.1$ | $4.7 \pm 0.1$ | $6.8 \pm 0.1$ |
| Split-MNIST | EWC | $0.9 \pm 0.0$ | $4.2 \pm 0.1$ | $0.9 \pm 0.1$ | $2.5 \pm 0.1$ |
| | VI | $17.6 \pm 0.4$ | $19.4 \pm 0.3$ | $5.1 \pm 0.3$ | $7.5 \pm 0.5$ |
| Split-CIFAR10 | EWC | $34.4 \pm 0.2$ | $47.9 \pm 0.5$ | $34.7 \pm 0.9$ | $39.7 \pm 1.4$ |
| | VI | $49.8 \pm 0.1$ | $52.2 \pm 0.2$ | $49.5 \pm 0.1$ | $52.5 \pm 0.2$ |

Table 2: Average and final cumulative error percentage and forgetting for Split-ImageNet on 3 random seeds. Lower is better, a random model achieves $98\%$ cumulative error.

| Domain | Method | CuL | Bound (equation 3) | Forgetting |
|--------|--------|-----|--------------------|------------|
| Split-ImageNet | EWC | $33.5 \pm 0.2$ | $40.1 \pm 0.3$ | $5.7 \pm 0.4$ |
| | SGD | $34.8 \pm 0.3$ | $41.4 \pm 0.4$ | $7.8 \pm 0.1$ |
| | Replay | $55.7 \pm 0.7$ | $62.5 \pm 0.4$ | $2.8 \pm 0.1$ |

## 5.1 VISION-BASED TASKS

In order to examine the bounds of Corollary 3.1 and Theorem 3.2, we made use of a few well-known computer vision tasks in the context of continual learning, namely: (1) Permuted-MNIST (Goodfellow et al., 2015), a domain-incremental problem (De Lange et al., 2021) where a random permutation is applied on each image for each task. (2) Split-MNIST (Zenke et al., 2017), a sequential set of binary classification tasks constructed from the MNIST (LeCun et al., 1998) dataset. (3) Split-CIFAR10 (Zenke et al., 2017), a sequential set of binary classification tasks constructed from the CIFAR-10 (Krizhevsky et al., 2009) dataset. (4) Split-ImageNet, a sequential set of binary classification tasks constructed from the ImageNet (Deng et al., 2009) dataset.

We used both variational inference (VI) (Hoffman et al., 2013) and Elastic Weight Consolidation (EWC) (Kirkpatrick et al., 2017) in our experiments. For Split-ImageNet, we also examined SGD with and without experience replay. Detail on the prior and posterior distributions used is available in Appendix B. We note that there is no significant computational or memory overhead required for bound calculation and no additional samples are required besides the training set.

We used convolutional neural networks (CNNs), with $T = 120$ tasks in total for all domains. We measured the average cumulative error $\frac{1}{T}\mathrm{CuL}((Q_t^\lambda)_{t=1}^T)$, approximated via a held-out test set for each task, and its upper bound based on equation 3 across 5 random seeds, reporting standard error and the average value. We also report the loss and upper bound for the final task. A full detailing of hyper-parameters and the experimental setup is available in Appendix B.

Tables 1, 2 detail the error percentage (error out of $100\%$, similar to accuracy percentage) for both VI and EWC models on all tasks, as well as the cumulative error and average test forgetting (see definition of forgetting measure (FM) in Wang et al. (2024a)) as well as Appendix B) for Split-ImageNet. The values reported are for both the average cumulative error and the error at the stopping point. Figure 3 shows the average cumulative loss and the upper bound as a function of the number of tasks for the split-MNIST task. We note that the values reported in the first two columns of Table 1 correspond to the final values of Figure 3, and that the last two columns correspond to only the error and bound for the final task in the continual learning process. Looking over the results, we see that the bound is very tight for VI, and somewhat looser for EWC. This is to be expected, to an extent, as the VI algorithm aims to optimize the r.h.s. of equation 3 w.r.t. the posterior distributions, whereas for EWC the bound is not directly linked to the learning process. Across all settings, we see that the bound becomes increasingly tight as the number of tasks increases, owing to both the fact that several terms in equation 3 decrease in proportion to the number of tasks, and to the tendency of the KL-divergence term to decrease during the continual learning process - as we experience additional tasks, the posterior for the previous task $Q_{t-1}$ that serves as the prior for task $\mathcal{D}_t$ becomes an increasingly better informed and predictive prior, resulting in a tighter upper bound. With the exception of the VI algorithm for the split-CIFAR10 task, all of our empirical upper bounds are non-vacuous, with several of them being tight enough to provide a useful risk certificate, especially if we consider only later tasks. The upper bounds for average cumulative error tend to suffer for

early tasks, possibly due to the randomness of the training process for the first few tasks. This is somewhat encouraging within the context of continually retraining a complex model to handle new data, as the forward transfer would be weighted towards generalizing on new tasks, and our bounds are consistently tighter (empirically) for later tasks in a sequence.

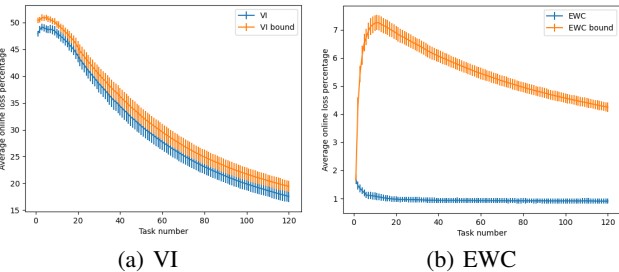

(a) VI        (b) EWC

Figure 3: Average CuL percentage and upper bound over time for split-MNIST. Error bars represent standard error over 5 random seeds. A random predictor would have $50\%$ average loss.

## 5.2 ORACLE BOUNDS

While the oracle bounds (4, 5) are already useful, as can be seen in the explicit oracle bounds for specific scenarios, verifying and comparing the bounds empirically may provide additional insights. To that end, we considered equation 4 and equation 5 in several simple scenarios that correspond to the specific theoretical scenarios discussed in Section 4. We utilized linear regression tasks following a similar setup as discussed in Lin et al. (2023) (a linear ground truth (Belkin et al., 2018; Evron et al., 2022) with the true weight vector for each task being a subset of all features) while varying the model between over and under-parametrized linear regression, and simple 2 layer fully connected (FC) neural networks (with a wide hidden layer to approximately adhere to the NTK regime (Jacot et al., 2018; Bennani & Sugiyama, 2020)). In order to approximate posterior sampling from the Gibbs posterior, we used the Stochastic Gradient Langevin Dynamic (SGLD) (Neal, 2011; Welling & Teh, 2011) algorithm in our experiments. The full detail of task construction and model and training hyper-parameters is available in Appendix B, alongside detailed numeric results.

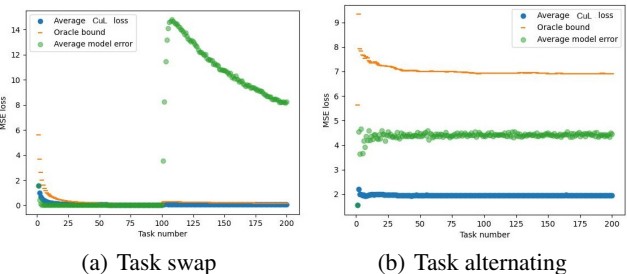

(a) Task swap        (b) Task alternating

Figure 4: Average cumulative loss, oracle bound (equation 4) and average model error over time for under-parametrized linear regression. (a) task changes at $t = 100$ (b) tasks alternate.

Figure 4 compares scenarios 2 and 3 for under-parametrized linear regression, with each task characterized by a different true weight vector. We can clearly see that task order matters significantly for both the cumulative loss and the average model loss. Specifically, we see that for cumulative loss, a single swap is significantly better than alternating between two tasks. This result agrees with the obtained theoretical bound. As can be seen for average model loss from the expected behavior detailed in Lin et al. (2023) (that agrees with the empirical average model loss for most of the continual learning process), a sudden task swap results in a sudden significant increase in average error that is slowly corrected, whereas alternating tasks quickly stabilize to a constant average error.

Figure 5 contrasts over and under-parametrized 2 layer neural networks on a sequence of gradually changing tasks corresponding to scenario 4 of Section 4. Each task corresponds to a generating linear weight vector, and adjacent tasks have similar weight vectors (see Appendix B). We can see

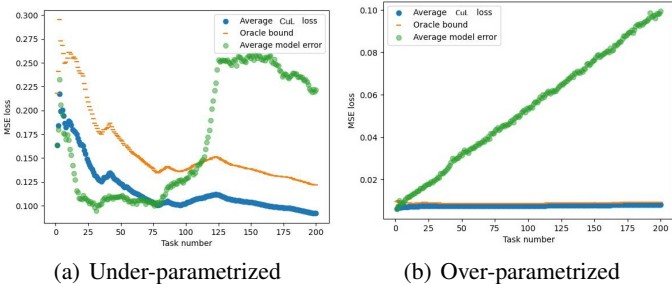

Figure 5: CuL, oracle bound (equation 4) and average model error over time for linear regression with gradually changing tasks. (a) Under-parametrized model (b) over-parametrized model.

that in the over-parametrized regime, cumulative error remains near constant and the upper bound is nearly exact. In the under-parametrized regime, however, we see a gradual decrease as the model stabilizes. In both cases, the average error tends to increase with the number of tasks (though it is lower in the over-parametrized setting), likely due to the increasing distance between tasks resulting from the random walk process. We note that our bounds are several orders of magnitude tighter in this setting compared to NTK-based generalization bounds for SGD (Bennani & Sugiyama, 2020).

## 6 LIMITATIONS AND FUTURE WORK

In this work, we derived several upper bounds on the cumulative error for both general hypothesis classes and for the Gibbs posterior. Our bounds offer tight risk certificates when the number of samples per task is large as well as in several concrete scenarios for the oracle bounds. Our results assume that the loss is either bounded or is sub-Gaussian, though extensions to heavy-tailed losses similarly to Haddouche & Guedj (2023) may be possible. The assumption of a strict global minimum can be relaxed to allow for a finite number of global minima. We note that our derived oracle bounds are taken w.r.t. the expected Gibbs posterior. Equivalent bounds can be derived w.r.t. empirical Gibbs posterior by modifying our assumptions on the global minimum to apply in expectation.

While our results are applicable for both offline and online continual learning, we acknowledge that for offline continual learning the cumulative error is often less relevant than the average error. A common assumption in this setting is an unbounded number of training samples per task, making learning plasticity irrelevant as any task can be learned from scratch. We also note that in the online continual learning setting, if task boundaries are blurry or unknown, the number of samples per task $m$ and the number of tasks $T$ must be approximated in order to use our bounds.

Our PAC-Bayes bounds contain complexity terms (KL-divergence) that may be difficult to scale for large models with many parameters. While this is not an issue in our oracle bounds, it is a concern for the general bounds such as equation 5. While there is some work in the context of PAC-Bayes bounds with other divergence measures (Bégin et al., 2016; Amit et al., 2022; Kuzborskij et al., 2024), this can be a potential limiting factor in applying our results for large model classes, though model-compression bounds (Lotfi et al., 2022) may serve as an avenue to overcome this limit.

We tested and verified our theoretical results on two main algorithms for several simple computer vision benchmarks, yielding non-vacuous bounds on the cumulative test error. As we can see in Tables 1, 2, the risk certificate is not always tight. We note that in most cases, the VI bound is nearly tight whereas the upper bounds for deterministic methods tend to be looser. This is somewhat unsurprising as the VI training objective attempts to directly optimize the r.h.s. of the upper bound.. As our main focus was measuring the efficacy of our upper bounds, we focused mostly on relatively small neural networks and classification problems. While encouraging, our preliminary experiments only used vision datasets, and we only examined our bounds on VI methods, EWC and experience replay algorithms. A more comprehensive empirical analysis of common continual learning methods, combined with applying some of the recent insights into obtaining tight risk certificates via PAC-Bayes bounds (Pérez-Ortiz et al., 2021), may yield further insights into the practical application of PAC-Bayes bounds for cumulative loss in continual learning for larger models.

## REPRODUCIBILITY STATEMENT

Proofs for all Theorems and Corollaries is available in Appendix A. All required assumptions appear in both the main text and the Appendix. Code for reproducing the experiments is available as part of the supplementary material and a more detailed explanation of the experimental setting and hyper-parameters is available in Appendix B. All data sources are publicly available and code is available on github.

## ACKNOWLEDGMENTS

This work was partially supported by grant 1693/22 from the Israel Science Foundation, and by the Skillman chair in biomedical sciences (RM)

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

# A PROOFS

**Theorem A.1.** *(Haddouche & Guedj (2022), Theorem 2.3) Assuming $\forall h \in \mathcal{H}, z \in \mathcal{Z}, \ell(h, z) \in [0, K]$, for online learning ($m = 1$), for any distribution $\mu$ over tasks $[T]$, any $\lambda > 0$ and any online predictive sequence $(P_t)$, for any sequence of stochastic kernels $(Q_t)$, we have with probability at least $1 - \delta$ over the sample $S = (z_1, \ldots, z_t) \sim \mu$ the following, holding for the data-dependent measures $Q_{t,S} \triangleq Q_t(S, \cdot), P_{t,S} \triangleq P_t(S, \cdot)$,*

$$\sum_{t=1}^{T} \mathbb{E}_{h_t \sim Q_{t,S}}[\mathbb{E}_{z_t \sim \mu}[\ell(h_t, z_t)|\mathcal{F}_{t-1}]] \leq \sum_{t=1}^{T} \left( \mathbb{E}_{h_t \sim Q_{t,S}}[\ell(h_t, z_t)] + \frac{1}{\lambda} \mathrm{KL}(Q_{t,S}||P_{t,S}) \right)$$
$$+ \frac{\lambda T K^2}{2} + \frac{\log(1/\delta)}{\lambda}.$$

**Corollary A.2.** *Restatement of Corollary 3.1: Under Assumption 1, for any $\lambda > 0$, for any set of $T$ tasks, for any online predictive sequence of priors $(P_t)$, for any sequence of stochastic kernels $(Q_t)$ with probability at least $1 - \delta$ over the draw of $(S_1, \ldots, S_T) \sim \mathcal{D}_1^m \times \ldots \times \mathcal{D}_T^m$, we have that the following holds for the data-dependent measures $P_t(S_{1:t}, \cdot), Q_t(S_{1:t}, \cdot)$,*

$$\frac{1}{T} \sum_{t=1}^{T} [\mathcal{L}(Q_t(S_{1:t}), \mathcal{D}_t)|\mathcal{F}_{t-1}] \leq \frac{1}{T} \sum_{t=1}^{T} \hat{\mathcal{L}}(Q_t(S_{1:t}), S_t) + \frac{1}{\lambda T} \sum_{t=1}^{T} \mathrm{KL}(Q_t(S_{1:t})||P_t(S_{1:t}))$$
$$+ \frac{\lambda K^2}{m} + \frac{\log(1/\delta)}{\lambda T}$$

*Proof.* We begin by applying the main Theorem of Rivasplata et al. (2020) with $Q^0 = P_1 \otimes \ldots \otimes P_T, \quad Q = Q_1 \otimes \ldots \otimes Q_T$, and with

$$F(S = S_1 \otimes \ldots \otimes S_T, h = h_1 \otimes \ldots \otimes h_T) = \frac{\lambda'}{T} \left( \sum_{t=1}^{T} \mathbb{E}_{z_t \sim \mathcal{D}_t}[\ell(h_t, z_t)|\mathcal{F}_{t-1}] - \sum_{t=1}^{T} \hat{\mathcal{L}}(h_t, S_t) \right).$$

Reorganizing terms, we have with probability at least $1 - \delta$ (over the draw of $S = S_1 \otimes \ldots \otimes S_T$),

$$\frac{1}{T} \sum_{t=1}^{T} \mathbb{E}_{h_t \sim Q_t(S_{1:t})}[\mathbb{E}_{z_t \sim \mathcal{D}_t}[\ell(h_t, z_t)|\mathcal{F}_{t-1}]] \leq \frac{1}{T} \sum_{t=1}^{T} \hat{\mathcal{L}}(Q_t(S_{1:t}), S_t)$$
$$+ \frac{1}{\lambda'} \sum_{t=1}^{T} \mathrm{KL}(Q_t(S_{1:t})||P_t(S_{1:t})) \qquad (6)$$
$$+ \frac{1}{\lambda'} \log \xi_T + \frac{\log(1/\delta)}{\lambda'}$$

where

$$\xi_T = \mathbb{E}_S \mathbb{E}_{h \sim Q^0} \left[ e^{F(S,h)} \right].$$

We then apply a similar Lemma D.2 of Haddouche & Guedj (2022) (with tasks instead of single examples) that yields

$$\xi_T = \prod_{t=1}^{T} \mathbb{E}_{S_1, \ldots, S_t} \mathbb{E}_{h_t \sim P_t} \left[ e^{\lambda'/T(\mathbb{E}_{z_t \sim \mathcal{D}_t}[\ell(h_t, z_t)|\mathcal{F}_{t-1}] - \hat{\mathcal{L}}(h_t, S_t))} \right],$$

and applying Hoeffding's Lemma for bounded losses or the exponential moment bound for sub-Gaussian random variables we have

$$\xi_T \leq e^{\frac{\lambda'^2 K^2}{mT}}.$$

Combined with equation 6, we have as follows:

$$\frac{1}{T} \sum_{t=1}^{T} \mathbb{E}_{h_t \sim Q_t(S_{1:t})}[\mathbb{E}_{z_t \sim \mathcal{D}_t}[\ell(h_t, z_t)|\mathcal{F}_{ti-1}]] \leq \frac{1}{T} \sum_{t=1}^{T} \hat{\mathcal{L}}(Q_t(S_{1:t}), S_i)$$
$$+ \frac{1}{\lambda'} \sum_{t=1}^{T} \mathrm{KL}(Q_t(S_{1:t})||P_t(S_{1:t}))$$
$$+ \frac{\lambda' K^2}{mT} + \frac{\log(1/\delta)}{\lambda'}$$

Picking $\lambda' = \lambda T$ completes the proof. □

**Theorem A.3.** *Restatement of Theorem 3.2: Under the same setup as Corollary 3.1, assuming $\ell(h, z) \in [0, K]$, for any predictive sequence of posteriors $(Q_t)$, for any $\delta_2 \in (0, 1]$, with probability at least $1 - \delta$ over the draw of $(S_1, \ldots, S_T) \sim \mathcal{D}_1^m \times \ldots \times \mathcal{D}_T^m$, the following holds for the data-dependent measures $P_t(S_{1:t}), Q_t(S_{1:t})$*

$$
\frac{1}{T} \sum_{t=1}^{T} [\mathcal{L}(Q_t(S_{1:t}), \mathcal{D}_t) | \mathcal{F}_{t-1}] \leq \frac{1}{T} \sum_{t=1}^{T} \hat{\mathcal{L}}(Q_t(S_{1:t}), S_t) + \frac{1}{\lambda} \sum_{t=1}^{T} \mathrm{KL}(Q_t(S_{1:t}) || P_t(S_{1:t}))
$$
$$
+ \frac{1}{\lambda} \log \left\{ (1 - \delta_2) e^{\lambda K \sqrt{\log(1/\delta_2)/2mT}} + \delta_2 e^{\lambda K} \right\}
$$
$$
+ \frac{\log(1/\delta)}{\lambda}
$$

*Proof.* Starting from equation 6 with

$$
F(S = S_1 \otimes \ldots \otimes S_T, h = h_1 \otimes \ldots \otimes h_T) = \lambda \left( \frac{1}{T} \sum_{t=1}^{T} \mathbb{E}_{z_i \sim \mathcal{D}_t}[\ell(h_t, z_t) | \mathcal{F}_{t-1}] - \frac{1}{T} \sum_{t=1}^{T} \hat{\mathcal{L}}(h_t, S_t) \right)
$$

for convenience, we have

$$
\frac{1}{T} \sum_{t=1}^{T} \mathbb{E}_{h_t \sim Q_t(S_{1:t})}[\mathbb{E}_{z_t \sim \mathcal{D}_t}[\ell(h_t, z_t) | \mathcal{F}_{t-1}]] \leq \frac{1}{T} \sum_{t=1}^{T} \hat{\mathcal{L}}(Q_t(S_{1:t}), S_t)
$$
$$
+ \frac{1}{\lambda} \sum_{t=1}^{T} \mathrm{KL}(Q_t(S_{1:t}) || P_t(S_{1:t}))
$$
$$
+ \frac{1}{\lambda} \log \xi_T + \frac{\log(1/\delta)}{\lambda}
$$

Since each element of $\frac{1}{mT} \sum_{t=1}^{T} \sum_{j=1}^{m} \ell(h, z_{tj})$ is bounded in range $[0, K/mT]$, we can apply Hoeffding's Lemma on each task; we first apply Markov's inequality: for $s, \epsilon > 0$

$$
\Pr(F(S = S_1 \otimes \ldots \otimes S_T, h = h_1 \otimes \ldots \otimes h_T) \geq \epsilon) = \Pr(e^{sF(S,h)} \geq e^{s\epsilon})
$$
$$
\leq e^{-s\epsilon} \mathbb{E}_{S,h} e^{s\lambda \left( \frac{1}{T} \sum_{t=1}^{T} \mathbb{E}_{z_t \sim \mathcal{D}_t}[\ell(h_t, z_t) | \mathcal{F}_{t-1}] - \frac{1}{T} \sum_{t=1}^{T} \hat{\mathcal{L}}(h_t, S_t) \right)}.
$$

Since we assume that the expected loss is $\mathcal{F}_{t-1}$-measurable, this equals

$$
= e^{-s\epsilon} \prod_{t=1}^{T} \mathbb{E}_{S_1, \ldots, S_t} \mathbb{E}_{h_1, \ldots, h_t} e^{s\lambda \left( \frac{1}{T} \mathbb{E}_{z_t \sim \mathcal{D}_t}[\ell(h_t, z_t)] - \frac{1}{mT} \sum_{i=1}^{m} \ell(h_t, z_{t,i}) \right)}.
$$

Since we assume data from each task is drawn **iid** we have

$$
= e^{-s\epsilon} \prod_{t=1}^{T} \mathbb{E}_{S_1, \ldots, S_{t-1}} \mathbb{E}_{h_1, \ldots, h_{t-1}} \prod_{i=1}^{m} \mathbb{E}_{S_t, h_t} e^{s\lambda \left( \frac{1}{mT} \mathbb{E}_{z_t \sim \mathcal{D}_t}[\ell(h_t, z_t)] - \frac{1}{mT} \ell(h_t, z_{t,i}) \right)}
$$
$$
\leq e^{-s\epsilon} \prod_{t=1}^{T} \prod_{i=1}^{m} e^{s^2(\lambda K/mT)^2/8}
$$
$$
= e^{\frac{s\lambda^2 K^2}{8mT\epsilon}}.
$$

Where the last inequality is due to applying Hoeffding's Lemma on each element in the product. Minimizing $s$, we have

$$\Pr(F(S = S_1 \otimes \ldots \otimes S_T, h = h_1 \otimes \ldots \otimes h_T) \geq \epsilon) \leq e^{-\frac{\epsilon^2 mT}{\lambda^2 K^2}} \triangleq \delta_2.$$

Moving terms around, we get

$$\epsilon = \lambda K \sqrt{\frac{\log(1/\delta_2)}{2mT}}.$$

We can then split $\xi_T = \mathbb{E}_S \mathbb{E}_{h \sim Q^0} \left[ e^{F(S,h)} \right]$ into two events with appropriate probabilities, with one event (the good event, with probability $1 - \delta_2$) fulfilling this inequality and the other (the bad event, with probability at most $\delta_2$) violating it giving us an upper limit $F(S, h) \leq \lambda K$, giving us

$$\xi_T \leq (1 - \delta_2) e^{\lambda K \sqrt{\log(1/\delta_2)/2mT}} + \delta_2 e^{\lambda K}.$$

Plugging in this inequality in equation 6 completes the proof. $\qquad\square$

**Lemma A.4.** *Under the same setting as Theorem 3.1, assuming that*

1. $Q_t = \hat{Q}_t^\lambda$ *is the empirical Gibbs measure*

2. $P_1 = P$ *is a data-free measure*

3. $\forall t > 1 : P_t = Q_{t-1} = \hat{Q}_{t-1}^\lambda$

*we have*

$$\mathbb{E}_{S_1,\ldots,S_T} \frac{1}{T} \sum_{t=1}^{T} [\mathcal{L}(\hat{Q}_t^\lambda(S_{1:t}), \mathcal{D}_t) | \mathcal{F}_{t-1}] \leq \mathbb{E}_{S_1,\ldots,S_T} \frac{1}{T} \sum_{t=1}^{T} \hat{\mathcal{L}}(\hat{Q}_{t-1}^\lambda(S_{1:t-1}), S_t) + \frac{\lambda K^2}{m} \qquad (7)$$

*Proof.* Starting from Theorem 3.1 (in expectation), we begin by decomposing the KL-divergence under our assumptions:

$$\frac{1}{\lambda T} \mathrm{KL}(\hat{Q}_t^\lambda \| \hat{Q}_{t-1}^\lambda) = \frac{1}{\lambda T} \mathbb{E}_{h \sim \hat{Q}_t^\lambda} \left[ \log \frac{e^{-\lambda \hat{\mathcal{L}}(h, S_t)}}{\mathbb{E}_{h \sim \hat{Q}_{t-1}^\lambda} e^{-\lambda \hat{\mathcal{L}}(h, S_t)}} \right] \qquad (8)$$

$$= -\frac{1}{T} \hat{\mathcal{L}}(\hat{Q}_t^\lambda, S_t) - \frac{1}{\lambda T} \log \mathbb{E}_{h \sim \hat{Q}_{t-1}^\lambda} e^{-\lambda \hat{\mathcal{L}}(h, S_t)}$$

Applying this equality to Theorem 3.1, we have

$$\frac{1}{T} \sum_{t=1}^{T} \mathbb{E}_{S_1,\ldots,S_i} [\mathcal{L}(\hat{Q}_t^\lambda(S_{1:t}), \mathcal{D}_t) | \mathcal{F}_{t-1}] \leq -\frac{1}{\lambda T} \sum_{t=1}^{T} \mathbb{E}_{S_1,\ldots,S_t} \log \mathbb{E}_{h \sim \hat{Q}_{t-1}^\lambda} e^{-\lambda \hat{\mathcal{L}}(h, S_t)} + \frac{\lambda K^2}{m} \qquad (9)$$

Applying Jensen's inequality on

$$-\frac{1}{\lambda T} \log \mathbb{E}_{h \sim \hat{Q}_{t-1}^\lambda} e^{-\lambda \hat{\mathcal{L}}(h, S_t)} \leq \frac{1}{T} \hat{\mathcal{L}}(\hat{Q}_{t-1}^\lambda, S_t)$$

for all $t \in [1, T]$ completes the proof. $\qquad\square$

**Theorem A.5.** *Restatement of Theorem 4.1: For any $\lambda > 0$, assuming*

1. $Q_t = Q_t^\lambda$ *is the expected Gibbs measure*

2. $P_1 = P$ *is a data-free measure over $\mathcal{H}$*

3. $\forall t > 1 : P_t = Q_{t-1} = Q_{t-1}^\lambda$

4. $\mathcal{H}$ *is a compact, bounded subset of $\mathbb{R}^d$.*

5. $\forall t \in [2, T]$, *the total expected loss* $\sum_{i=1}^{t} \mathcal{L}(h, \mathcal{D}_i)$ *has a strict global minimum at* $h_{1:t}^*$ *and is twice continuously differentiable w.r.t.* $h$.

*,we have*

$$\lim_{m,T\to\infty} \frac{1}{T} \sum_{t=1}^{T} [\mathcal{L}(Q_t^\lambda, \mathcal{D}_t) | \mathcal{F}_{t-1}] \leq \lim_{T\to\infty} \frac{1}{T} \sum_{t=2}^{T} \mathcal{L}(h_{1:t-1}^*, \mathcal{D}_t)$$

*Proof.* We begin with the change-of-measure inequality; for any $\lambda > 0$ and any measurable function $f : \mathcal{H} \times \mathcal{D}^M \to \mathbb{R}$, for any prior and posterior $P, Q$, for any sample $S \sim \mathcal{D}^M$ a.s.

$$-\frac{1}{\lambda} \log \mathbb{E}_{h\sim P} e^{-\lambda f(h,S)} \leq \mathbb{E}_{h\sim Q(S)} f(h, S) + \frac{1}{\lambda} \mathrm{KL}(Q(S)\|P)$$

Taking an expectation over $S$ and using Jensen's inequality, we have

$$-\mathbb{E}_{S\sim\mathcal{D}^M} \mathbb{E}_{h\sim Q(S)} f(h, S) \leq \frac{1}{\lambda} \log \mathbb{E}_{S\sim\mathcal{D}^M} \mathbb{E}_{h\sim P} e^{-\lambda f(h,S)} + \frac{1}{\lambda} \mathbb{E}_{S\sim\mathcal{D}^M} \mathrm{KL}(Q(S)\|P)$$

Choosing $Q = Q_1^\lambda \otimes \ldots \otimes Q_T^\lambda, P = P \otimes Q_1^\lambda \otimes \ldots \otimes Q_{T-1}^\lambda$ as well as

$$f(S = S_1 \otimes \ldots \otimes S_T, h = h_1 \otimes \ldots \otimes h_T) = -\left( \frac{1}{T} \sum_{t=1}^{T} \mathbb{E}_{z_t\sim\mathcal{D}_t} [\ell(h_t, z_t) | \mathcal{F}_{t-1}] - \frac{1}{T} \sum_{t=1}^{T} \hat{\mathcal{L}}(h_t, S_t) \right)$$

yields

$$\frac{1}{T} \sum_{t=1}^{T} [\mathcal{L}(Q_t^\lambda, \mathcal{D}_t) | \mathcal{F}_{t-1}] \leq \frac{1}{T} \sum_{t=1}^{T} \mathbb{E}_{S_t\sim\mathcal{D}_t^m} [\hat{\mathcal{L}}(Q_t^\lambda, S_t) | \mathcal{F}_{t-1}]$$

$$+ \frac{1}{\lambda'} \log \mathbb{E}_S \mathbb{E}_{h_1,\ldots,h_T\sim P,,\ldots,Q_{T-1}^\lambda} e^{\lambda'/T(\sum_{t=1}^{T} (\mathbb{E}_{z_t\sim\mathcal{D}_t}[\ell(h_t,z_t)|\mathcal{F}_{t-1}] - \hat{\mathcal{L}}(h_t,S_t)))}$$

$$+ \frac{1}{\lambda'} \sum_{t=1}^{T} \mathrm{KL}(Q_t^\lambda \| Q_{t-1}^\lambda)$$

$$(10)$$

Using Hoeffding's Lemma for bounded losses or the exponential moment bound for sub-Gaussian random variables, we get

$$\frac{1}{T} \sum_{t=1}^{T} [\mathcal{L}(Q_t^\lambda, \mathcal{D}_t) | \mathcal{F}_{t-1}] \leq \frac{1}{T} \sum_{t=1}^{T} \mathbb{E}_{S_t\sim\mathcal{D}_t^m} [\hat{\mathcal{L}}(Q_t^\lambda, S_t) | \mathcal{F}_{t-1}] + \frac{\lambda' K^2}{mT} + \frac{1}{\lambda'} \sum_{t=1}^{T} \mathrm{KL}(Q_t^\lambda \| Q_{t-1}^\lambda)$$

Picking $\lambda' = \lambda T$ and decomposing the KL-divergence as before, we get

$$\frac{1}{T} \sum_{t=1}^{T} [\mathcal{L}(Q_t^\lambda, \mathcal{D}_t) | \mathcal{F}_{t-1}] \leq \frac{1}{T} \sum_{t=1}^{T} \left( \mathbb{E}_{S_t\sim\mathcal{D}_t^m} [\hat{\mathcal{L}}(Q_t^\lambda, S_t) - \mathcal{L}(Q_t^\lambda, \mathcal{D}_t) | \mathcal{F}_{t-1}] \right)$$

$$+ \frac{\lambda K^2}{m} - \frac{1}{\lambda T} \sum_{t=1}^{T} \log \mathbb{E}_{h\sim Q_{t-1}^\lambda} e^{-\lambda \mathcal{L}(h,\mathcal{D}_t)} \qquad (11)$$

by unrolling the last term according to the definition of the Gibbs posterior, i.e.

$$\mathbb{E}_{h\sim Q_{t-1}^\lambda} e^{-\lambda \mathcal{L}(h,\mathcal{D}_t)} = \frac{\mathbb{E}_{h\sim Q_{t-2}^\lambda} e^{-\lambda \mathcal{L}(h,\mathcal{D}_t) - \lambda \mathcal{L}(h,\mathcal{D}_{t-1})}}{\mathbb{E}_{h\sim Q_{t-2}^\lambda} e^{-\lambda \mathcal{L}(h,\mathcal{D}_{t-1})}}$$

$$= \ldots = \frac{\mathbb{E}_{h\sim P} e^{-\lambda \sum_{j=1}^{t-1} \mathcal{L}(h,\mathcal{D}_j)}}{\mathbb{E}_{h\sim P} e^{-\lambda \sum_{j=1}^{t-1} \mathcal{L}(h,\mathcal{D}_j) - \lambda \mathcal{L}(h,\mathcal{D}_t)}},$$

we get

$$\frac{1}{T}\sum_{t=1}^{T}[\mathcal{L}(Q_t^\lambda, \mathcal{D}_t)|\mathcal{F}_{t-1}] \leq \frac{1}{T}\sum_{t=1}^{T}\left(\mathbb{E}_{S_t\sim\mathcal{D}_t^m}[\hat{\mathcal{L}}(Q_t^\lambda, S_t) - \mathcal{L}(Q_t^\lambda, \mathcal{D}_t)|\mathcal{F}_{t-1}]\right)$$

$$+ \frac{\lambda K^2}{m} - \frac{1}{\lambda T}\log\mathbb{E}_{h\sim P^\lambda}e^{-\lambda\mathcal{L}(h,\mathcal{D}_1)} \qquad (12)$$

$$+ \frac{1}{\lambda T}\sum_{t=2}^{T}\log\frac{\mathbb{E}_{h\sim P}e^{-\lambda\sum_{j=1}^{t-1}\mathcal{L}(h,\mathcal{D}_j)}}{\mathbb{E}_{h\sim P}e^{-\lambda\sum_{j=1}^{t-1}\mathcal{L}(h,\mathcal{D}_j)-\lambda\mathcal{L}(h,\mathcal{D}_t)}}$$

Suppose that $\mathcal{H}$ is a compact, bounded subset of $\mathbb{R}^d$. Assuming that $\mathcal{L}(h,\mathcal{D}_j)$ is twice continuously differentiable w.r.t. $h$ for all $j$, we can apply Laplace's method (Shun & McCullagh, 1995) on both numerator and denominator. Let

$$M_{1:t-1}(h) \triangleq \frac{1}{i-1}\sum_{j=1}^{t-1}\mathcal{L}(h,\mathcal{D}_j),$$

$$h_{1:t-1}^* \triangleq \arg\min_{h\in\mathcal{H}} M_{1:t-1}(h) = \arg\min_{h\in\mathcal{H}}\sum_{j=1}^{t-1}\mathcal{L}(h,\mathcal{D}_j)$$

, then marking $M_{1:t-1}'' = \det M_{1:t-1}''(h_{1:t-1}^*)|$ the determinant of the Hessian matrix, we have the Taylor expansion

$$\frac{1}{\lambda T}\sum_{t=2}^{T}\log\frac{\mathbb{E}_{h\sim P}e^{-\lambda\sum_{j=1}^{t-1}\mathcal{L}(h,\mathcal{D}_j)}}{\mathbb{E}_{h\sim P}e^{-\lambda\sum_{j=1}^{t-1}\mathcal{L}(h,\mathcal{D}_j)-\lambda\mathcal{L}(h,\mathcal{D}_t)}} =$$

$$\frac{1}{\lambda T}\sum_{t=2}^{T}\log\frac{\left(\frac{2\pi}{\lambda(t-1)}\right)^{d/2}\frac{1}{\sqrt{M_{1:t-1}''}}e^{-\lambda\sum_{j=1}^{t-1}\mathcal{L}(h_{1:t-1}^*,\mathcal{D}_j)} \quad R}{\left(\frac{2\pi}{\lambda(t-1)}\right)^{d/2}\frac{1}{\sqrt{M_{1:t-1}''}}e^{-\lambda\sum_{j=1}^{t-1}\mathcal{L}(h_{1:t-1}^*,\mathcal{D}_j)-\lambda\mathcal{L}(h_{1:t-1}^*,\mathcal{D}_t)} \quad R}$$

where $R = 1 + \frac{J_2}{2\lambda(t-1)M_{1:t-1}''} + \ldots + O((\lambda(t-1))^{-r-1})$.

Due to setting the same function to seek an optimum for in both the numerator and denominator (with differing reminder). Since most of the elements of both numerator and denominator are the same, we have (in the limit where $\lambda\to\infty$)

$$\frac{1}{\lambda T}\sum_{t=2}^{T}\log\frac{\mathbb{E}_{h\sim P}e^{-\lambda\sum_{j=1}^{t-1}\mathcal{L}(h,\mathcal{D}_j)}}{\mathbb{E}_{h\sim P}e^{-\lambda\sum_{j=1}^{t-1}\mathcal{L}(h,\mathcal{D}_j)-\lambda\mathcal{L}(h,\mathcal{D}_t)}} = \frac{1}{\lambda T}\sum_{t=2}^{T}\log e^{\lambda\mathcal{L}(h_{1:t-1}^*,\mathcal{D}_i)} = \frac{1}{T}\sum_{t=2}^{T}\mathcal{L}(h_{1:t-1}^*,\mathcal{D}_t)$$

combined with equation 12 we have (in the limit)

$$\frac{1}{T}\sum_{t=1}^{T}[\mathcal{L}(Q_t^\lambda, \mathcal{D}_t)|\mathcal{F}_{t-1}] \leq \frac{1}{T}\sum_{t=1}^{T}\left(\mathbb{E}_{S_t\sim\mathcal{D}_t^m}[\hat{\mathcal{L}}(Q_t^\lambda, S_t) - \mathcal{L}(Q_t^\lambda, \mathcal{D}_t)|\mathcal{F}_{t-1}]\right)$$

$$+ \frac{\lambda K^2}{m} - \frac{1}{\lambda T}\log\mathbb{E}_{h\sim P^\lambda}e^{-\lambda\mathcal{L}(h,\mathcal{D}_1)} + \frac{1}{T}\sum_{t=2}^{T}\mathcal{L}(h_{1:t-1}^*,\mathcal{D}_t) \qquad (13)$$

Taking $\lambda = \sqrt{m/T}$, we have

$$\lim_{m,T\to\infty}\frac{1}{T}\sum_{t=1}^{T}[\mathcal{L}(Q_t^\lambda, \mathcal{D}_t)|\mathcal{F}_{t-1}] \leq \lim_{T\to\infty}\frac{1}{T}\sum_{t=2}^{T}\mathcal{L}(h_{1:t-1}^*,\mathcal{D}_t) + \lim_{T\to\infty}\frac{1}{T}\mathcal{L}(P,\mathcal{D}_1)$$

$$\square$$

**Corollary A.6.** *Restatement of Corollary 4.2: For any $\lambda > 0$, assuming*

  *1. $Q_t = Q_t^\lambda$ is the expected Gibbs measure*

2. $P_1 = P$ is a data-free measure over $\mathcal{H}$

3. $\forall t > 1 : P_t = Q_{t-1} = Q_{t-1}^\lambda$

4. $\mathcal{H}$ is a compact, bounded subset of $\mathbb{R}^d$.

5. $\forall t \in [2, T]$, the expected loss $\mathcal{L}(h, \mathcal{D}_{t-1})$ has a strict global minimum at $h_{t-1}^*$ and is twice continuously differentiable w.r.t. $h$.

*we have*

$$\lim_{m \to \infty} \frac{1}{T} \sum_{t=1}^{T} [\mathcal{L}(Q_t^\lambda, \mathcal{D}_t) | \mathcal{F}_{t-1}] \le \frac{1}{T} \sum_{t=2}^{T} \mathcal{L}(h_{t-1}^*, \mathcal{D}_t) + \frac{1}{T} \mathcal{L}(P, \mathcal{D}_1)$$

*Proof.* Via unrolling the last term in equation 11 once we get

$$
\begin{aligned}
\frac{1}{T} \sum_{t=1}^{T} [\mathcal{L}(Q_i^\lambda, \mathcal{D}_t) | \mathcal{F}_{t-1}] \le &\frac{1}{T} \sum_{t=1}^{T} \left( \mathbb{E}_{S_t \sim \mathcal{D}_t^m} [\hat{\mathcal{L}}(Q_t^\lambda, S_t) - \mathcal{L}(Q_t^\lambda, \mathcal{D}_t) | \mathcal{F}_{t-1}] \right) \\
&+ \frac{\lambda K^2}{m} - \frac{1}{\lambda T} \log \mathbb{E}_{h \sim P} e^{-\lambda \mathcal{L}(h, \mathcal{D}_1)} \\
&+ \frac{1}{\lambda T} \sum_{t=2}^{T} \log \frac{\mathbb{E}_{h \sim Q_{t-2}^\lambda} e^{-\lambda \mathcal{L}(h, \mathcal{D}_{t-1})}}{\mathbb{E}_{h \sim Q_{t-2}^\lambda} e^{-\lambda \mathcal{L}(h, \mathcal{D}_{t-1}) - \lambda \mathcal{L}(h, \mathcal{D}_t)}}
\end{aligned}
\tag{14}
$$

Using similar arguments of Laplace's approximation, we get for $\lambda = \sqrt{m}$,

$$\lim_{m \to \infty} \frac{1}{T} \sum_{t=1}^{T} [\mathcal{L}(Q_i^\lambda, \mathcal{D}_t) | \mathcal{F}_{t-1}] \le \frac{1}{T} \sum_{t=2}^{T} \mathcal{L}(h_{t-1}^*, \mathcal{D}_t) + \frac{1}{T} \mathcal{L}(P, \mathcal{D}_1)$$

$\square$

**Corollary A.7.** *Under the same conditions as Theorem 4.1, if we also have that*

$$\forall i, j, h \in \mathcal{H}, \quad |\mathcal{L}(h, \mathcal{D}_i) - \mathcal{L}(h, \mathcal{D}_j)| \le \mathcal{G}_\mathcal{H} d(\mathcal{D}_i, \mathcal{D}_j)$$

*for some non-negative $\mathcal{G}_\mathcal{H}$ and metric $d(\cdot, \cdot)$, then*

$$\lim_{m, T \to \infty} \frac{1}{T} \sum_{t=1}^{T} [\mathcal{L}(Q_t^\lambda, \mathcal{D}_t) | \mathcal{F}_{t-1}]$$

$$\le \lim_{m, T \to \infty} \left[ \frac{1}{T} \sum_{t=2}^{T} \frac{1}{t-1} \min_{h \in \mathcal{H}} \sum_{j=1}^{t-1} \mathcal{L}(h, \mathcal{D}_j) + \frac{\mathcal{G}_\mathcal{H}}{T} \sum_{t=2}^{T} \frac{1}{t-1} \sum_{j=1}^{t-1} d(\mathcal{D}_j, \mathcal{D}_t) + O\left(\frac{1}{T}\right) \right]$$

where the $O\left(\frac{1}{T}\right)$ term is $\frac{1}{T} \mathcal{L}(P, \mathcal{D}_1)$.

*Proof.* Starting from equation 13, we decompose

$$\mathcal{L}(h_{1:t-1}^*, \mathcal{D}_t) = \frac{t-1}{t-1} \mathcal{L}(h_{1:t-1}^*, \mathcal{D}_t) + \frac{1}{t-1} \sum_{j=1}^{t-1} \mathcal{L}(h_{1:t-1}^*, \mathcal{D}_j) - \frac{1}{t-1} \sum_{j=1}^{t-1} \mathcal{L}(h_{1:t-1}^*, \mathcal{D}_j)$$

$$= \frac{1}{t-1} \sum_{j=1}^{t-1} \mathcal{L}(h_{1:t-1}^*, \mathcal{D}_j) + \frac{1}{t-1} \sum_{j=1}^{t-1} \left( \mathcal{L}(h_{1:t-1}^*, \mathcal{D}_t) - \mathcal{L}(h_{1:t-1}^*, \mathcal{D}_j) \right)$$

$$\le \frac{1}{t-1} \sum_{j=1}^{t-1} \mathcal{L}(h_{1:t-1}^*, \mathcal{D}_j) + \frac{1}{t-1} \sum_{j=1}^{t-1} \left| \mathcal{L}(h_{1:t-1}^*, \mathcal{D}_t) - \mathcal{L}(h_{1:t-1}^*, \mathcal{D}_j) \right|$$

By the definition of $h_{1:t-1}^*$,

$$= \frac{1}{t-1} \min_{h \in \mathcal{H}} \sum_{j=1}^{t-1} \mathcal{L}(h, \mathcal{D}_j) + \frac{1}{t-1} \sum_{j=1}^{t-1} \left| \mathcal{L}(h_{1:t-1}^*, \mathcal{D}_t) - \mathcal{L}(h_{1:t-1}^*, \mathcal{D}_j) \right|$$

$$\leq \frac{1}{t-1} \min_{h \in \mathcal{H}} \sum_{j=1}^{t-1} \mathcal{L}(h, \mathcal{D}_j) + \frac{\mathcal{G}_{\mathcal{H}}}{t-1} \sum_{j=1}^{t-1} d(\mathcal{D}_t, \mathcal{D}_j) \qquad (15)$$

Using equation 15 to upper bound each loss in equation 13 and taking the limit completes the proof.

□

# B    Appendix - empirical setting and hyper-parameters

All experiments were run on local hardware with an NVIDIA GeForce 4090 GPU and an Intel i9 CPU. All results were run for 5 random seeds and averages and standard error were reported in all tables. Anonymized code is available in the supplementary material and full code on github.

## B.1    Pseudo-code for bound calculation

---
**Algorithm 1** Continual learning training and bound calculation for VI

---
**function** CONTINUAL-LEARN($S_1, \ldots, S_T, P$)
    Choose algorithmic parameters $\lambda_1, \ldots, \lambda_T$
    Let $\hat{Q}_{1:0}(h) \triangleq P(h)$
    **for** each task $t$ from 1 to $T$ **do**
        Update $\hat{Q}_{1:t}$ via

$$\hat{Q}_{1:t} = \arg\min_{Q_{1:t}} \left\{ \hat{\mathcal{L}}(Q_{1:t}, S_t) + \frac{1}{\lambda_t} D_{\text{KL}}(Q_{1:t} || \hat{Q}_{1:t-1}) \right\}$$

        Update upper bound for (test) CuL based on equation 1 or 3
    **return** $\hat{Q}_{1:T}$, upper bound

---

---
**Algorithm 2** Continual learning training and bound calculation for deterministic methods

---
**function** CONTINUAL-LEARN($S_1, \ldots, S_T, P$)
    Choose algorithmic hyper-parameters $\theta$, bound parameters $\lambda_1, \ldots, \lambda_T, \sigma$
    initialize model weights $w_0$
    Let $\hat{Q}_{1:0} \triangleq \mathcal{N}(w_0, \sigma^2 I_d)$
    **for** each task $t$ from 1 to $T$ **do**
        Update model weights $w_t$ via the algorithm $\mathcal{A}_\theta(w_0, \ldots, w_{t-1}, S_t)$
        Let $\hat{Q}_{1:t} =\triangleq \mathcal{N}(w_t, \sigma^2 I_d)$
        Update upper bound for (test) CuL based on equation 1 or 3.
    **return** $w_T$, upper bound

---

## B.2    Posterior construction

For the variational inference (VI) (Hoffman et al., 2013) algorithm, both the prior and posterior distributions were multivariate Gaussian distributions on model parameters.

For deterministic methods such as EWC, a posterior distribution is constructed after training by adding Multivariate Gaussian noise, i.e. $\hat{Q}_{1:t} = \mathcal{N}(w_t, \sigma^2 I_d)$, where $w_t \in \mathcal{R}^d$ is the weight vector given as output after task $\mathcal{D}_t$.

## B.3    Vision datasets

For all vision tasks except Split-ImageNet, we used a convolutional neural network consisting of convolution blocks each consisting of 64 two-dimensional convolutions, max-pooling and tanh activations. The convolution blocks are then followed by a fully connected layer and an additional tanh activation. Reported results also use a linear classification head for each task, but results without separate classification heads per task were not significantly different. In all cases, training for a task consisted of a single pass over all training examples, and expected error for a task is estimated via a held-out test set.

For both MNIST-based tasks, we used two convolution blocks of $5 \times 5$ convolutions and the linear layer contained 400 neurons. For the CIFAR10-based task, we used three convolution blocks of $3 \times 3$ convolutions and the linear layer contained 800 neurons. For Split-ImageNet, we used a pre-trained ResNet-18 (He et al., 2016) model.

For permuted-MNIST, we used a different pixel permutation per task, and each task involved 10-way classification. All 60000 training samples were used for training with a batch size of 128. The

learning rate was static at $1e^{-3}$ and the $\lambda$ parameter was set to $1e^{-5}$. For split-MNIST, each task involved half of the labels (at random) chosen as positive and half as negative. This is a minor departure from the standard split-MNIST problem where 5 different binary classification tasks are created and their loss is averaged, but the overall behavior is similar. All other hyper-parameters were set identically to permuted-MNIST. For split-CIFAR10, task construction was the same as split-MNIST but on the CIFAR10 dataset. All 50000 training samples were used for training with a batch size of 256. The learning rate was static at $1e^{-3}$ and the $\lambda$ parameter was set to $5e^{-4}$. For split-ImageNet, each task contained a disjoint subset of 50 classes, and the $\lambda$ parameter was set to $1e^{-7}$. The notion of forgetting measured and reported for split-ImageNet is the average test forgetting, defined as

$$FM(h_t, \mathcal{D}_1, \ldots, \mathcal{D}_{t-1}) = \frac{1}{t-1} \sum_{j=1}^{t-1} \max_{i \in [1,t-1]} \{a(h_i, \mathcal{D}_j) - a(h_t, \mathcal{D}_j)\},$$

where $a(h, \mathcal{D}_t)$ is the test accuracy on task $t$. This measure can be written in terms of the loss function as

$$FM(h_t, \mathcal{D}_1, \ldots, \mathcal{D}_{t-1}) = \frac{1}{t-1} \sum_{j=1}^{t-1} \max_{i \in [1,t-1]} \{\ell(h_t, \mathcal{D}_j) - \ell(h_i, \mathcal{D}_j)\}.$$

For VI, we used Markov Chain Monte Carlo (MCMC) estimation with 3 draws from the posterior, and the prior was a multivariate Gaussian with noise $\sigma_0^2 = 5e^{-2}$ and posterior noise $\sigma_t^2 = 1e^{-4}$. For EWC, the $\sigma^2$ parameter used for posterior construction was set as $1e^{-4}$, and the regularization weight was set at $\lambda_{\mathrm{EWC}} = 100$ for all datasets except Split-ImageNet, and $\lambda_{\mathrm{EWC}} = 40$ for Split-ImageNet. For experience replay, a replay buffer of size 1000 was used. Training was done using the Adam optimizer (Kingma & Ba, 2015) except for Split-ImageNet, where SGD with weight decay of $1e^{-4}$ was used. Hyper-parameters were chosen via manual trial and error using a held-out validation set.

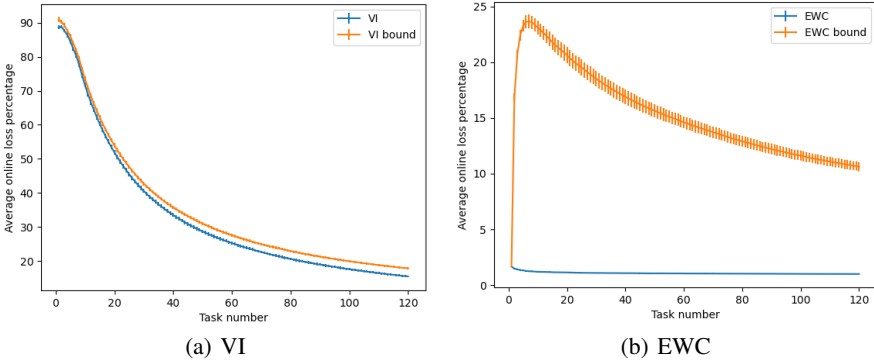

(a) VI                                            (b) EWC

Figure 6: Average cumulative loss percentage and upper bound over time for permuted-MNIST. Error bars represent standard error over 5 random seeds. A random predictor would have $50\%$ average cumulative loss.

### B.4 ORACLE BOUND EXPERIMENTS

Tasks were constructed as linear regression tasks of the form $Y_t = X_t^T w_t^* + \epsilon_t$, where each element in $X_t$ follows standard Gaussian distribution $N(0,1)$, and $\epsilon_t \sim N(0, \sigma^2 I_d)$ with $\sigma = 0.3$. Test data is drawn without noise $Y_t = X_t^T w_t^*$. All elements in $w_t^*$ follows standard Gaussian distribution $N(0,1)$. Like in Lin et al. (2023), for the linear case the true weight vector for each task $w_t^*$ is partially zeroed out (we zero out $80\%$ of the weights) to construct zero-filled features for different tasks. For non-linear experiments we use all input dimensions.

For the task swap setting, we use the same weight $w_1^*$ until $t = 100$, then swap to tasks with weight $w_2^*$ until $t = T = 200$. For the alternating task setting, we alternate between $w_1^*$ and $w_2^*$. For the

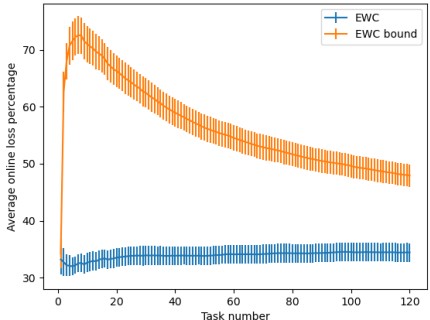

Figure 7: Average cumulative loss percentage and upper bound over time for split-CIFAR10 and the EWC algorithm. Error bars represent standard error over 5 random seeds. A random predictor would have $50\%$ average cumulative loss.

gradual change setting, we draw an initial normalized $w_1^*$ ($||w_1^*||_2^2 = 1$), and for any new task we update $w_{t+1}^* = (w_t^* + \epsilon)/||w_t^* + \epsilon||_2^2$, where $\epsilon_t \sim N(0, (0.3)^2 I_d)$.

We ran SGLD (Apache 2.0) for 20 iterations on each task, with an initial temperature of $3e^{-3}$ and halving temperature after each epoch. The learning rate was constant at $1e^{-3}$. We generated 2048 training samples per task and 400 test samples. The training batch size for SGLD was fixed at 128. We used a total of $T = 200$ tasks for all settings. Loss is measured via the mean square error (MSE).

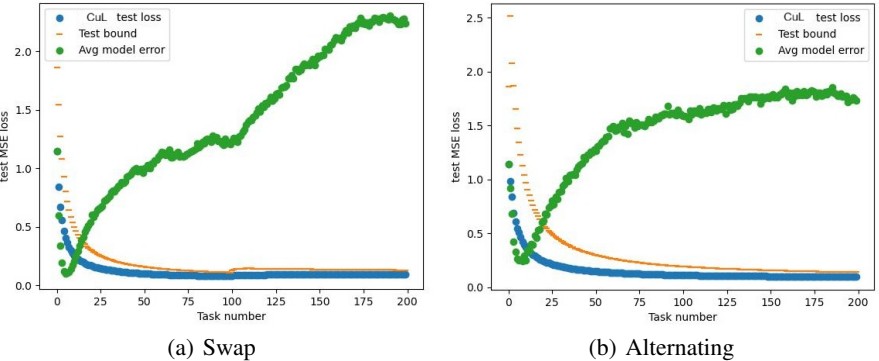

(a) Swap            (b) Alternating

Figure 8: Average cumulative loss, oracle bound equation 4 and average model error over time for linear regression with Under-parametrized non-linear models. (a) Task swap (b) Alternating tasks. Cumulative loss and the oracle bound converge to similar values.

For linear experiments, we used $w_t^* \in \mathbb{R}^{600}$ and $x_t \in \mathbb{R}^{3000}$. For non-linear (deep) experiments, we used $w_t^* \in \mathbb{R}^{10}$, using a fully connected neural network with one hidden layer with ReLU activations. For the under-parametrized experiments, the hidden layer was of dimension 100, and for the over-parametrized experiments, the hidden layer was of dimension 4000.

## C   LLM USAGE

LLMs were used during the editing process of the paper for punctuation and checking for grammatical errors.

Table 3: Average and final cumulative error (MSE) percentage for oracle datasets. Lower is better.

| Domain | Method | Cumulative error | Bound (equation 4) | Average error |
|---|---|---|---|---|
| Swap | Linear over-parametrized | $10.3 \pm 0.2$ | $19.0 \pm 0.3$ | $572.7 \pm 10.8$ |
| Swap | Deep under-parametrized | $0.11 \pm 0.01$ | $0.15 \pm 0.01$ | $2.72 \pm 0.47$ |
| Swap | Deep over-parametrized | $0.01 \pm 0.00$ | $0.01 \pm 0.00$ | $0.12 \pm 0.00$ |
| Alternating | Linear over-parametrized | $280.1 \pm 5.4$ | $520.8 \pm 10.3$ | $398.3 \pm 8.1$ |
| Alternating | Deep under-parametrized | $0.11 \pm 0.01$ | $0.16 \pm 0.01$ | $2.02 \pm 0.21$ |
| Alternating | Deep over-parametrized | $0.01 \pm 0.00$ | $0.01 \pm 0.00$ | $0.12 \pm 0.00$ |
| Gradual | Linear over-parametrized | $4.7 \pm 0.2$ | $8.2 \pm 0.4$ | $7.8 \pm 1.1$ |
| Gradual | Deep under-parametrized | $0.10 \pm 0.00$ | $0.12 \pm 0.01$ | $0.19 \pm 0.02$ |
| Gradual | Deep over-parametrized | $0.01 \pm 0.00$ | $0.01 \pm 0.00$ | $0.10 \pm 0.00$ |

