# OpenReview forum: "PAC-Bayes bounds for cumulative loss in Continual Learning"
_ICLR.cc/2026/Conference — ICLR 2026 Poster_

### Official Review · Reviewer_9vg7 · 2025-10-30

**Soundness:** 2
**Presentation:** 3
**Contribution:** 2
**Rating:** 4
**Confidence:** 4

**Summary:**

The paper extends existing PAC-Bayes bounds used for online and offline learning to address the setting of continual learning, where tasks arrive sequentially. The authors focus on deriving general upper bounds on cumulative generalization loss applicable to any task distribution and learning algorithm. The paper also addresses oracle bounds for Gibbs posteriors and compares these approaches across various task distributions, as well as testing their empirical validity on vision-based continual learning problems. This work represents the effort to establish upper bounds on learning plasticity within the domain of continual learning, given its unique challenge of balancing knowledge retention and forward transfer.

**Strengths:**

1. **Rigorous Theory**: The extension of PAC-Bayes bounds to continual learning, especially in providing non-vacuous bounds, is a theoretical contribution.

2. **Empirical Validation for the Theory**: The authors provide empirical evidence that supports their theoretical claims across various task distributions, including vision-based datasets. Experiments on linear regression tasks demonstrate the effectiveness of their bounds.

3. **Good Presentation:** The paper is well written and easy to follow.

**Weaknesses:**

1. **Use of Cumulative Loss:** The Cumulative Loss is a typical measure in online/continual meta-learning. For continual learning, the bound should be analyzed with the average loss in Definition 4. In contrast, the whole paper focused on cumulative loss.

2. **Novelty and Related Work Discussion:** The paper claims to be the ﬁrst general upper bound on learning plasticity for continual learning, while it missed a lot of theoretical work in continual learning [1, 2, 3], and its most related work is in online/continual meta-learning [4,5,6,7]. In particular, the discussion regarding [4, 5, 6, 7] should be in depth, where these methods also brought novel algorithms to reduce the cumulative loss as well as improve its upper bounds.

3. **Theoretical Limitation:** The proposed bounds depend on $\mathcal{G}_{\mathcal{H}} d(\mathcal{D}_1, \mathcal{D}_2)$, which are hard to evaluate and the bound could be very loose. In addition, the results require that the number of samples per task should exceed the square root of the total number of tasks, which neglects the positive transfer among tasks.

4. **Limited Experiments:** The experiments are somehow too simple.

[1] Itay Evron, Edward Moroshko, Rachel Ward, Nathan Srebro, and Daniel Soudry. How catastrophic can catastrophic forgetting be in linear regression? In Conference on Learning Theory, pp. 4028–4079. PMLR, 2022.

[2] Hongbo Li, Sen Lin, Lingjie Duan, Yingbin Liang, and Ness Shroff. Theory on mixture-of-experts in continual learning. In The Thirteenth International Conference on Learning Representations, 2025.

[3] Sen Lin, Peizhong Ju, Yingbin Liang, and Ness Shroff. Theory on forgetting and generalization of continual learning. In International Conference on Machine Learning, pp. 21078–21100. PMLR, 2023.

[4] Giulia Denevi, Carlo Ciliberto, Riccardo Grazzi, and Massimiliano Pontil. Learning-to-learn stochastic gradient descent with biased regularization. In International Conference on Machine Learning, pages 1566–1575. PMLR, 2019.

[5] Qi Chen, Changjian Shui, Ligong Han, and Mario Marchand. On the stability-plasticity dilemma in continual meta-learning: Theory and algorithm. Advances in Neural Information Processing Systems, 36:27414–27468, 2023.

[6] Maria-Florina Balcan, Mikhail Khodak, and Ameet Talwalkar. Provable guarantees for gradientbased meta-learning. In International Conference on Machine Learning, pages 424–433. PMLR, 2019.

[7] Mikhail Khodak, Maria-Florina Balcan, and Ameet Talwalkar. Adaptive gradient-based metalearning methods. arXiv preprint arXiv:1906.02717, 2019.

**Questions:**

As the authors have acknowledged several limitations in the main paper, and considering the questions raised above, I would like to know how the authors plan to address them.

I would expect the authors to adequately resolve most of my concerns; otherwise, I would be reluctant to recommend acceptance, as the limitations appear to be quite significant.

---

> ### Author Response · Authors · 2025-11-20
>
> Thank you for your thorough and insightful review.
>
> **Regarding the cumulative loss**, as we discuss in the general comment, while we certainly agree that the average loss is a useful and important metric in the context of continual learning, the cumulative loss is a measure of learning plasticity. While several theoretical papers exist that explore forgetting and average loss in the context of continual learning exist such as [1,2,3,10, 11], the cumulative loss is far less explored in this setting. As we can see from several recent empirical papers [12,13,14], both average and cumulative loss are important for effective continual learning. Cumulative loss is especially relevant to online continual learning (see [13]) where we are only allowed a single pass on the data for each task.
>
> **Regarding related work and discussions of novelty**, we do cite [1, 3], though we agree with your and other reviewer opinions that the related work section should be extended. We would like to note that most existing work in continual learning theory such as [1,2,3] measure forgetting and the average error. Regarding [4], we note that we do discuss meta-transfer risk in reference to [8,9]. [5,6,7] all discuss regret (which is indeed related to cumulative loss) in the context of meta-learning, where training data from previous tasks remains available. We will include a more comprehensive comparison of our work to existing settings in the revised version. We also note that existing upper bounds on cumulative loss in continual learning are specific to optimization algorithms and/or problem structures.
>
> **Regarding theoretical limitations**, as mentioned in our response to other reviews and in the limitations section, some of our assumptions may be relaxed in future work. While the assumption of bounded/sub-Gaussian loss is non-trivial, it is relatively common to PAC-Bayes bounds and may be possible to relax to heavy-tailed losses such as in [15]. While the assumption of a strict global minimum or a finite set of strict global minima does somewhat limit the applicability of our oracle bounds, other oracle bounds with different sets of assumptions can be derived from Theorem 3.2. Assumption 2 (Lipchitz loss) is a fairly strong assumption used for several specific examples and applications of the more general Theorem 4.1. While estimating Lipchitz constants may be difficult in some settings, the use of Lipchitz assumptions is helpful in giving concrete and easier to understand examples for the more general results. Only Equation 3 requires that the number of samples per task should exceed the square root of the total number of tasks ($m>\\sqrt{T}$) to converge as $m,T\\rightarrow \\infty$. You raise a valid point in stating that the general result does not explicitly take advantage of any positive transfer between tasks. Implicitly, if tasks exhibit strong positive transfer, we can choose a lower value for $\\lambda$ in Equation 1, resulting in a tighter overall bound.
>
> **Regarding the experimental section**, as we note in the limitations section, our empirical results are somewhat preliminary, as the main focus of the paper was theoretical. Still, we examine several optimization algorithms, vision datasets and architectures, as well as consider some of the settings for oracle bounds. We would welcome a discussion on what would be considered a more complete or comprehensive set of experiments in your opinion. There is still some time until a revised version is to be submitted, and additional experiments can be conducted.

---

> > ### Author Response · Authors · 2025-11-20
> >
> > [1] Itay Evron, Edward Moroshko, Rachel Ward, Nathan Srebro, and Daniel Soudry. How catastrophic can catastrophic forgetting be in linear regression? In Conference on Learning Theory, pp. 4028–4079. PMLR, 2022.
> >
> > [2] Hongbo Li, Sen Lin, Lingjie Duan, Yingbin Liang, and Ness Shroff. Theory on mixture-of-experts in continual learning. In The Thirteenth International Conference on Learning Representations, 2025.
> >
> > [3] Sen Lin, Peizhong Ju, Yingbin Liang, and Ness Shroff. Theory on forgetting and generalization of continual learning. In International Conference on Machine Learning, pp. 21078–21100. PMLR, 2023.
> >
> > [4] Giulia Denevi, Carlo Ciliberto, Riccardo Grazzi, and Massimiliano Pontil. Learning-to-learn stochastic gradient descent with biased regularization. In International Conference on Machine Learning, pages 1566–1575. PMLR, 2019.
> >
> > [5] Qi Chen, Changjian Shui, Ligong Han, and Mario Marchand. On the stability-plasticity dilemma in continual meta-learning: Theory and algorithm. Advances in Neural Information Processing Systems, 36:27414–27468, 2023.
> >
> > [6] Maria-Florina Balcan, Mikhail Khodak, and Ameet Talwalkar. Provable guarantees for gradientbased meta-learning. In International Conference on Machine Learning, pages 424–433. PMLR, 2019.
> >
> > [7] Mikhail Khodak, Maria-Florina Balcan, and Ameet Talwalkar. Adaptive gradient-based metalearning methods. arXiv preprint arXiv:1906.02717, 2019.
> >
> > [8] Anastasia Pentina and Christoph H Lampert. Lifelong learning with non-iid tasks. Advances in Neural Information Processing Systems, 28, 2015.
> >
> > [9] Ron Amit and Ron Meir. Meta-learning by adjusting priors based on extended pac-bayes theory. In International Conference on Machine Learning, pp. 205–214. PMLR, 2018.
> >
> > [10] Lior Friedman and Ron Meir. Data-dependent and oracle bounds on forgetting in continual learning. In Proceedings of The 4nd Conference on Lifelong Learning Agents, 2025.
> >
> > [11] Thang Doan, Mehdi Abbana Bennani, Bogdan Mazoure, Guillaume Rabusseau, and Pierre Alquier. A theoretical analysis of catastrophic forgetting through the ntk overlap matrix. In International Conference on Artificial Intelligence and Statistics, pp. 1072–1080. PMLR, 2021.
> >
> > [12] Dohare, S., Hernandez-Garcia, J. F., Lan, Q., Rahman, P., Mahmood, A. R., & Sutton, R. S. (2024). Loss of plasticity in deep continual learning. Nature, 632(8026), 768-774.
> >
> > [13] Wang, M., Michel, N., Xiao, L., & Yamasaki, T. (2024). Improving plasticity in online continual learning via collaborative learning. In Proceedings of the IEEE/CVF Conference on Computer Vision and Pattern Recognition (pp. 23460-23469).
> >
> > [14] Kumar, S., Marklund, H., & Van Roy, B. (2025, February). Maintaining Plasticity in Continual Learning via Regenerative Regularization. In Conference on Lifelong Learning Agents (pp. 410-430). PMLR.
> >
> > [15] Maxime Haddouche and Benjamin Guedj. Pac-bayes generalisation bounds for heavy-tailed losses through supermartingales. Transactions on Machine Learning Research, 2023.

---

### Official Review · Reviewer_cjXg · 2025-11-02

**Soundness:** 2
**Presentation:** 2
**Contribution:** 2
**Rating:** 6
**Confidence:** 3

**Summary:**

The paper provides a PAC-Bayes bound on the cumulative loss (loss summed over each retraining of the model) for continual learning, which is an evaluation metric that measures the efficacy of learning algorithms over an entire predictive sequence. The author then demonstrates the efficacy of the proposed bounds on standard continual learning datasets.

**Strengths:**

The paper is clearly written as an extension of previous papers, namely (Haddouche & Guedj, 2022). The writing and proofs are clear and seem technically correct. The formulation of the problem is original as the consideration of CuL is indeed new.

**Weaknesses:**

Minor:
1. Although the presentation is clear, the writing can be more inviting to readers. E.g, some brief introduction on the setup of PAC-Bayes bounds in general can greatly help the readers with the required background.
2. The related work section can include some newer papers, e.g [1,2], that propose both empirical methods, but also with the appropriate theoretical analysis on forgetting/knowledge transfer bounds.

Major:
1. Some of the assumptions might be unrealistic.
2. The bounds seem loose in scenarios where a proper continual learning technique was applied in the setting.

[1]: Wu, Yichen, Long-Kai Huang, Renzhen Wang, Deyu Meng, and Ying Wei. "Meta continual learning revisited: Implicitly enhancing online hessian approximation via variance reduction." In The Twelfth international conference on learning representations, vol. 2. 2024.
[2]: Yang, Haoming, Ali Hasan, and Vahid Tarokh. "Parabolic Continual Learning." In International Conference on Artificial Intelligence and Statistics, pp. 2620-2628. PMLR, 2025.

**Questions:**

1. In Figure 1, the author used the notation $Q_{1:t}$, but it doesn't seem like this notation is used anywhere else in the analysis. Is this notation the same as $Qt$?
2. Line 187-188, Can the author address why, in a continual learning context, the upper bounds that converge as the number of tasks increases are vastly preferable? This is an important motivation to motivate the rest of the analysis in the paper.
3. Line 189-190, Is the assumption that the loss function is upper-bounded by a constant realistic? Most of the loss functions, such as MSE and cross-entropy, are not upper-bounded. The author should provide a few applicable loss functions here as an example.
4.  In the case of equation 3, can the author explain which part of the bound applying a proper continual learning method reduces? Intuitively, does a buffer-based algorithm help reduce the first term, while a regularization-based algorithm reduces the KL term of the bounds?
5. In the experiments, error percentage is used to evaluate the efficacy of different methods as a loss function, but it is generally not used to optimize a learning algorithm. Will this violate the assumptions made to prove the bounds?

---

> ### Author Response · Authors · 2025-11-20
>
> Thank you for your positive and informative review.
>
> **Regarding the introduction**, we thank you for your suggestions regarding improving the readability of our results and the paper overall. We originally included a more detailed introduction connecting our results to [1] but shortened it to fit page count requirements. Your review as well as others clearly indicate that we can improve our introduction and comparison, and we will do so in the revised version.
>
> **Regarding related work**, we mainly focused on theoretical work as we consider that a more direct comparison. There are many good empirical methods, some of which like the suggested papers with theoretical groundings. We would like to point out, however, that these works mainly focus on forgetting or the average error. Results that aim to tackle plasticity measures from a theoretically grounded perspective, however, are far fewer.
>
> **Regarding our assumptions**, as we state in our limitations section, some of the assumptions may be possible to relax in future work. Our general bound makes assumptions about the loss function that are typical for PAC-Bayes bounds, and makes no strong assumptions regarding the task space (e.g. independent tasks, strong task similarity). Our oracle bounds make stronger assumptions that do not necessarily hold for all cases, but are slightly less restrictive compared to strict convexity assumptions. Other oracle bounds can be derived from Theorem 3.2 by adding additional assumptions such as a finite hypothesis space or a restricted hypothesis class.
>
> **Regarding the looseness of the bounds for deterministic algorithms** such as EWC and experience replay, as we mention in the paper (e.g. lines 366-369) this is not necessarily surprising, as the upper bound of Equation 3 is general, and is applied post-hoc to a deterministic method via a relatively simple method that is agnostic to the training process. Many techniques to improve upper bounds of this nature such as learning the prior [6] can be applied to the continual learning setting to improve the tightness of the bounds in practice.
>
> **As for your specific questions**:
> 1. Thank you for your observation, you are correct. We agree that this should be better clarified, and we will change $Q\_{1:t}$ to $Q\_t$ for the revised version.
> 2.	The continual learning setting is one where we assume that we will encounter many related tasks in a sequence. Ideally, we would want an upper bound that converges as the number of data (examples per task times the number of tasks) increases. As such, while results such as Corollary 3.1 and Theorem 2.3 of [1] are useful, we would prefer bounds that converge as the total number of task increases even if the number of samples per task does not. We will endeavour to better clarify this in the revised version.
>
> 3.	Note that we assume that the loss used for the bound calculation is either upper bounded by a constant or sub-Gaussian (low-tailed). Some relevant examples are 0-1 classification loss, bounded cross-entropy (via a minimal probability) and MSE in some settings without outliers. We also note in the limitations section that it may be possible to extend our results to heavy-tailed losses similarly to [2]. We will add clarifying examples to the revised version as per your suggestion.
>
> 4.	Equation 3 is a general upper bound applicable regardless of the optimization algorithm used. Since most classic continual learning methods focus on forgetting rather than plasticity, mechanisms such as replay buffers and regularization both serve to reduce the KL term of the bound. In works focused on stability measures such as [3, 4, 5] a buffer-based algorithm would likely improve the empirical loss term while (memory-free) regularization-based methods attempt to reduce the loss for the current task and KL divergence terms.
>
> 5.	The bounds are applicable regardless of the optimization method, including the training loss. There is no issue with using an unbounded loss such as cross-entropy during training and classification loss for the upper bound. For variational inference (VI) we would prefer to use the same loss for training and the bound because VI is theoretically motivated by optimizing the r.h.s. of Equation 3, but the bounds themselves are applicable for any posterior distribution.

---

> > ### Author Response · Authors · 2025-11-20
> >
> > [1] Maxime Haddouche and Benjamin Guedj. Online pac-bayes learning. Advances in Neural Information Processing Systems, 35:25725–25738, 2022.
> >
> > [2] Maxime Haddouche and Benjamin Guedj. Pac-bayes generalisation bounds for heavy-tailed losses through supermartingales. Transactions on Machine Learning Research, 2023.
> >
> > [3] Lior Friedman and Ron Meir. Data-dependent and oracle bounds on forgetting in continual learning. In Proceedings of The 4nd Conference on Lifelong Learning Agents, 2025.
> >
> > [4] Anastasia Pentina and Christoph H Lampert. Lifelong learning with non-iid tasks. Advances in Neural Information Processing Systems, 28, 2015.
> >
> > [5] Sen Lin, Peizhong Ju, Yingbin Liang, and Ness B. Shroff. Theory on forgetting and generalization of continual learning. In International Conference on Machine Learning, ICML 2023, 23-29 July 2023, Honolulu, Hawaii, USA, volume 202 of Proceedings of Machine Learning Research, pp. 21078–21100. PMLR, 2023.
> >
> > [6] Maria Perez-Ortiz, Omar Rivasplata, John Shawe-Taylor, and Csaba Szepesvari. Tighter risk certificates for neural networks. Journal of Machine Learning Research, 22(227):1–40, 2021.

---

### Official Review · Reviewer_a1bF · 2025-11-02

**Soundness:** 3
**Presentation:** 1
**Contribution:** 3
**Rating:** 6
**Confidence:** 3

**Summary:**

The paper extends PAC-Bayes tools from online/time-uniform settings to derive upper bounds on the cumulative (forward) loss in continual learning. The authors give conditions under which the bounds converge as the number of tasks grows (notably when the per-task sample size scales sufficiently with the number of tasks), and they analyze several controlled scenarios (task repeats, alternation, gradual change) to interpret the oracle bounds. There is some empirical validations of the claims.

**Strengths:**

- The paper provides an original contribution by formulating PAC-Bayes bounds on the cumulative error in continual learning, addressing a problem that has received limited theoretical treatment so far.
- The specialization of the results to different continual learning scenarios (repeated tasks, alternating tasks, gradual changes) is a useful aspect that makes the theoretical framework more interpretable.
- The proof techniques might be of methodological interest.

**Weaknesses:**

* **Lack of narrative and intuition.**
  The paper is presented as a sequence of results with limited discussion. The authors should provide more intuition after each main theorem, explain the meaning of the key terms, and discuss when the bounds are informative.

* **Insufficient comparison with prior work.**
  The relationship with existing results (e.g., Friedman & Meir; Haddouche & Guedj) is not clearly established. A more explicit contrast between this setting and previous PAC-Bayes formulations for continual or online learning is needed.

* **Assumptions and applicability.**
  Some assumptions (bounded or sub-Gaussian loss, compact hypothesis space, strict minima) are strong and not discussed in detail. The paper should clarify how these assumptions affect the applicability of the results to neural networks.

* **Presentation and readability.**
  The paper assumes substantial familiarity with PAC-Bayes theory. Proof ideas and key steps should be summarized in the main text. Figures and tables could be improved for clarity.

* **Experimental section.**
  The experiments are not clearly described. Some results (e.g., high CIFAR10 error) are unexplained. It is unclear whether models were trained to convergence, and how optimization error is handled in the evaluation of bounds. Also, the visualizations are not well annotated.

* **Discussion of overparameterization.**
  Although the text mentions that overparameterization affects the bounds, this is not explored in detail. A more systematic analysis or clearer interpretation would improve the contribution.

**Questions:**

1. In Corollary 3.1, the constant (K) (and the assumption of bounded loss) should be re-introduced.
2. Why is the KL divergence measured only with respect to task-shared parameters (line 182)? Would it not be possible to include task-specific parameters by defining them as zero for other tasks?
3. The empirical results on CIFAR10 show unusually high errors. Could the authors clarify the training procedure and whether models were trained to convergence?
4. How is optimization error accounted for in the experiments? If models are not trained to convergence, how does this affect the interpretation of the bounds?
5. Could the authors explain why variational inference yields tighter bounds than EWC or SGD, beyond the observation that it optimizes a PAC-Bayes objective?
6. The oracle bounds assume strict global minima and compact hypothesis space. Are these assumptions essential, or can the results be extended to more realistic cases?

---

> ### Author Response · Authors · 2025-11-20
>
> Thank you for your comprehensive and informative review.
>
> In the paper itself we have attempted to give some intuition regarding each Theorem and its key steps and link them together in a cohesive narrative. We thank you for your suggestions regarding improving the readability of our results and the paper overall. We originally included a more detailed introduction connecting our results to [1] but shortened it to fit page count requirements. Your review as well as others clearly indicate that we can improve our introduction and comparison, and we will do so in the revised version.
>
> Your suggestion regarding clarifying and extending our comparison to previous work on PAC-Bayes bounds for average error in continual learning [2], online learning [1] and meta-learning [3] is appreciated, and we intend to extend this comparison given that the final version allows for an additional page.
>
> **Regarding our Assumptions**, as we mention in our limitations section, our results assume that the loss is either bounded or is sub-Gaussian, though extensions to heavy-tailed losses similarly to [5] may be possible. The assumption of a strict global minimum for oracle bounds can be relaxed to allow for a finite number of global minima. Our assumptions on the loss may limit the applicability of our results in some cases (e.g. regression with unbounded heavy-tailed losses). Our assumption of compact subsets of $\\mathbb{R}^d$ is rarely an issue in practice, as most neural networks use parameter spaces in the space of real numbers and digital representations are bounded. The assumption of strict global minima may or may not be an issue in practice depending on the model, the loss function and the task space, and our oracle bounds (unlike the more general upper bounds) only apply when this assumption holds.
>
> **Regarding the experimental section**, the experimental setup including hyper-parameters, architecture and optimization process is detailed in full in Appendix B. Seeing as you also added specific questions regarding this topic, we provide further details regarding each question. Models are not trained to convergence for each task but rather are trained on a single pass, but seeing as our bounds are generalization bounds, the optimization error is incorporated into the posterior distribution, so a sub-optimal optimization process yields a posterior with higher empirical loss.
>
> **Regarding overparameterization** and its impact on our results, this distinction is relevant in the context of the oracle bounds, where Assumption 3 details the sufficient condition assumed for overparameterization to meaningfully impact the cumulative error. In two specific settings (alternating tasks and task switching), we compare the optimal cumulative loss to the upper bound with sufficient overparameterization. Our experiments in Section 5.2 and Appendix B.3 reflect this difference in bound behavior. We will endeavor to better clarify this in the revised version.

---

> > ### Author Response · Authors · 2025-11-20
> >
> > **In response to your specific questions**:
> > 1. Corollary 3.1 clearly states that it is applicable under Assumption 1 (bounded/sub-Gaussian loss) that introduces the constant $K$ immediately before it. Since this Assumption is used in several of our results, we preferred to define it once for brevity.
> >
> > 2. In the case where model parameters are divided into shared parameters $Q\_{t}$ and separate task-specific parameters $q\_{t}$, the overall KL-divergence is $D\_{\\mathrm{KL}}(Q\_{t}||P\_{t})+ D_{\\mathrm{KL}}(q\_{t}||p\_{t})$. Due to the assumptions on measurability, we can choose $p\_t=q\_t$, removing the task-specific part of the divergence. This serves to lower the overall KL-divergence. You are correct that that task-specific parameters could be included in the bound calculation, and since parameters for other tasks do not change the KL-divergence is zero. Lines 181-185 state that in the case where both shared and task-specific parameters are part of the model, only the change in shared parameters is relevant to the bounds.
> >
> > 3. As we detail in Appendix B, results for split-CIFAR-10 used 3 blocks of 3x3 convolutions followed by max-pooling and tanh activations for the shared representation. Training for each task involved a single pass over all 50K training examples with a batch size of 256.
> >
> > 4. The optimization error is only relevant for non-oracle bounds. It is important to note that PAC-Bayes theory in general focuses mainly on the generalization error, and the errors are taken w.r.t. the posterior distribution. If the model has not converged or converged to a poor local minimum, the posterior distribution $Q\_{t}$ may be affected, potentially resulting in higher empirical and expected loss due to optimization error.
> >
> > 5. For deterministic algorithms (SGD, EWC, experience replay), the upper bound is calculated post-hoc after training using a relatively simple prior and posterior construction, so a higher KL-divergence term in the final bound is not surprising.  The variational inference algorithm explicitly optimizes the r.h.s. of Equation 1 or 3 w.r.t. each specific task, so any gap for the obtained posterior is small by virtue of the optimization process. Of note here is that while the upper bound for VI is tighter, the empirical cumulative error may be higher as in the results in Table 1.
> >
> > 6. Theorem 4.1 and its corollary use Laplace’s method and therefore require a compact hypothesis space that is a subset of $\\mathbb{R}^d$ as well as a strict global minimum. This result can be extended to a finite number of strict global minima (by choosing the one with the highest loss on task $t$). Other oracle bounds could potentially be derived from Theorem 3.2 via different assumptions such as for finite hypothesis spaces or for restricted model classes such as linear separators (see also [4] for an introduction to PAC-Bayes oracle bounds).
> >
> >
> > [1] Maxime Haddouche and Benjamin Guedj. Online pac-bayes learning. Advances in Neural Information Processing Systems, 35:25725–25738, 2022.
> >
> > [2] Lior Friedman and Ron Meir. Data-dependent and oracle bounds on forgetting in continual learning. In Proceedings of The 4nd Conference on Lifelong Learning Agents, 2025.
> >
> > [3] Ron Amit and Ron Meir. Meta-learning by adjusting priors based on extended pac-bayes theory. In International Conference on Machine Learning, pp. 205–214. PMLR, 2018.
> >
> > [4] Pierre Alquier. User-friendly introduction to pac-bayes bounds. Foundations and Trends® in Machine Learning, 17(2):174–303, 2024. ISSN 1935-8237. doi: 10.1561/2200000100.
> >
> > [5] Maxime Haddouche and Benjamin Guedj. Pac-bayes generalisation bounds for heavy-tailed losses through supermartingales. Transactions on Machine Learning Research, 2023.

---

### Author Response · Authors · 2025-11-20
**General comment on the cumulative loss and previous work**

Based on the reviews provided, we believe that there is room for general clarification regarding the stability-plasticity dilemma in continual learning and how our contribution relates to existing work (we intend to include this in our revision).

  A key element in continual learning is the trade-off between the ability to preserve performance on previous tasks (AKA maintaining memory stability or avoiding catastrophic forgetting) and the ability to learn new tasks effectively (AKA learning plasticity or forward transfer). Historically, early work in the domain of continual learning mainly focused on the issue of catastrophic forgetting and maintaining high memory stability, while recent work [1,2,3] has paid more attention to the subject of plasticity in continual learning.

Of the related theoretical work in the theory of continual learning, most such as [4,5,6,7] discuss the average loss or related measures of stability. Works focused on the neural tangent kernel (NTK) regime such as [8,9] do offer upper bounds on both cumulative and average loss, but are limited to specific optimization algorithms used during training and on the NTK regime itself.

We also note that the notion or regret that is sometimes discussed in both meta-learning (e.g. [10]) and online learning [11] is related and similar to the cumulative loss. These settings, however, differ from the continual learning setting in meaningful ways: meta-learning assumes continued access to previous tasks and often requires a shared distribution for tasks, and online learning pessimistically assumes that all samples are taken from different, possibly adversarial tasks. Accordingly, several common strategies from these fields such as online gradient descent and follow-the-leader algorithms are non-ideal or inapplicable to the continual learning domain.


[1] Dohare, S., Hernandez-Garcia, J. F., Lan, Q., Rahman, P., Mahmood, A. R., & Sutton, R. S. (2024). Loss of plasticity in deep continual learning. Nature, 632(8026), 768-774.

[2] Wang, M., Michel, N., Xiao, L., & Yamasaki, T. (2024). Improving plasticity in online continual learning via collaborative learning. In Proceedings of the IEEE/CVF Conference on Computer Vision and Pattern Recognition (pp. 23460-23469).

[3] Kumar, S., Marklund, H., & Van Roy, B. (2025, February). Maintaining Plasticity in Continual Learning via Regenerative Regularization. In Conference on Lifelong Learning Agents (pp. 410-430). PMLR.

[4] Itay Evron, Edward Moroshko, Rachel Ward, Nathan Srebro, and Daniel Soudry. How catastrophic can catastrophic forgetting be in linear regression? In Conference on Learning Theory, pp. 4028–4079. PMLR, 2022.

[5] Hongbo Li, Sen Lin, Lingjie Duan, Yingbin Liang, and Ness Shroff. Theory on mixture-of-experts in continual learning. In The Thirteenth International Conference on Learning Representations, 2025.

[6] Sen Lin, Peizhong Ju, Yingbin Liang, and Ness Shroff. Theory on forgetting and generalization of continual learning. In International Conference on Machine Learning, pp. 21078–21100. PMLR, 2023.

[7] Lior Friedman and Ron Meir. Data-dependent and oracle bounds on forgetting in continual learning. In Proceedings of The 4nd Conference on Lifelong Learning Agents, 2025.

[8] Bennani, M. A., & Sugiyama, M. Generalisation Guarantees for Continual Learning with Orthogonal Gradient Descent. In 4th Lifelong Machine Learning Workshop at ICML 2020.

[9] Thang Doan, Mehdi Abbana Bennani, Bogdan Mazoure, Guillaume Rabusseau, and Pierre Alquier. A theoretical analysis of catastrophic forgetting through the ntk overlap matrix. In International Conference on Artificial Intelligence and Statistics, pp. 1072–1080. PMLR, 2021.

[10] Qi Chen, Changjian Shui, Ligong Han, and Mario Marchand. On the stability-plasticity dilemma in continual meta-learning: Theory and algorithm. Advances in Neural Information Processing Systems, 36:27414–27468, 2023.

[11] Maxime Haddouche and Benjamin Guedj. Online pac-bayes learning. Advances in Neural Information Processing Systems, 35:25725–25738, 2022.

---

### Meta-Review · Area_Chair_5d7g · 2026-01-11

**Summary:**

This paper extends PAC-Bayes generalization theory to the continual learning (CL) setting by deriving upper bounds on cumulative generalization loss. Unlike prior work that focuses on average loss or specific algorithms, the authors present general bounds applicable to arbitrary task distributions and learning algorithms. They also introduce oracle bounds for Gibbs posteriors and analyze their behavior under various task dynamics (e.g., repeated, alternating, and gradually changing tasks). Empirical evaluations on vision datasets and linear regression tasks demonstrate that the proposed bounds are non-vacuous and interpretable.

**Reviewer Concerns:**

Reviewers appreciated the theoretical rigor and originality of the contribution but raised several concerns. Key issues included the lack of intuitive explanations and narrative structure, limited discussion of assumptions (e.g., bounded loss, compact hypothesis space), and insufficient comparison with related work in continual and meta-learning. Some reviewers questioned the practicality of the bounds due to strong assumptions and potential looseness, especially in realistic neural network settings. The experimental section was also critiqued for unclear setup, high error rates on CIFAR-10, and limited exploration of overparameterization effects.

**Reviewer Scores:**

The paper received mixed scores of 6, 6, and 4 from three reviewers. Reviewer a1bF rated it 6, acknowledging the originality of the theoretical contribution but noted weaknesses in presentation, limited intuition behind the results, and insufficient comparison with prior work. Reviewer cjXg also gave a score of 6, recognizing the clarity of the theoretical formulation but raised concerns about the realism of assumptions and the looseness of the bounds in practical settings. Reviewer 9vg7 rated the paper 4,having concerns on the use of cumulative loss, novelty and related works, loose bound, and simple experiments. The authors responsed to the questions on the relations to related works in online learning/meta-learning, clarified motivations on cumulative loss, and promised to revise and discuss in the final version.  Given the novelty of the theoretical contribution and the authors’ willingness to improve the clarity and contextualization of their work, the paper is a borderline accept case. A thorough revision that strengthens the related works, clarifies assumptions, and expands empirical and comparative analysis will be essential for the final version.

---

### Decision · Program_Chairs · 2026-01-26

Accept (Poster)